# Endothelial-specific FoxO1 depletion prevents obesity-related disorders by increasing vascular metabolism and growth

**Martina Rudnicki, Ghoncheh Abdifarkosh, Emmanuel Nwadozi, Sofhia V Ramos, Armin Makki, Diane M Sepa-Kishi, Rolando B Ceddia, Christopher GR Perry, Emilie Roudier, Tara L Haas\***

School of Kinesiology and Health Science and the Muscle Health Research Centre, York University, Toronto, Canada

**Abstract** Impaired angiogenesis is a hallmark of metabolically dysfunctional adipose tissue in obesity. However, the underlying mechanisms restricting angiogenesis within this context remain ill-defined. Here, we demonstrate that induced endothelial-specific depletion of the transcription factor Forkhead Box O1 (FoxO1) in male mice led to increased vascular density in adipose tissue. Upon high-fat diet feeding, endothelial cell FoxO1-deficient mice exhibited even greater vascular remodeling in the visceral adipose depot, which was paralleled with a healthier adipose tissue expansion, higher glucose tolerance and lower fasting glycemia concomitant with enhanced lactate levels. Mechanistically, FoxO1 depletion increased endothelial proliferative and glycolytic capacities by upregulating the expression of glycolytic markers, which may account for the improvements at the tissue level ultimately impacting whole-body glucose metabolism. Altogether, these findings reveal the pivotal role of FoxO1 in controlling endothelial metabolic and angiogenic adaptations in response to high-fat diet and a contribution of the endothelium to whole-body energy homeostasis.

DOI: https://doi.org/10.7554/eLife.39780.001

**\*For correspondence:**
thaas@yorku.ca

**Competing interests:** The authors declare that no competing interests exist.

## Introduction

Obesity is a growing problem worldwide (*Moller and Kaufman, 2005*; *Tchernof and Després, 2013*) and thus an urgent need exists to identify molecular processes and signaling pathways that may serve as novel therapeutic targets to hinder obesity-induced pathologies. Although the underlying causes of obesity-related complications are multifactorial, the dysfunction of adipose tissue plays a central role in the development of peripheral tissue metabolic disturbances, ultimately reflecting systemically in dyslipidemia, insulin resistance and hyperglycemia (*Moller and Kaufman, 2005*; *Fuster et al., 2016*).

Capillary endothelial cells (EC) are well-known regulators of tissue adaptation to pathologic challenges through their prominent role in blood vessel formation and remodeling. During expansion of visceral adipose tissue, impaired vascular remodeling promotes hypoxia, inflammation, and fibrosis (*Corvera and Gealekman, 2014*; *Fuster et al., 2016*). Conversely, forced stimulation of vascular growth in adipose tissue of obese rodents improves adipose tissue function (*Sun et al., 2012*; *Robciuc et al., 2016*; *Seki et al., 2018*), counteracting obesity-related metabolic disorders (*Sung et al., 2013*; *Seki et al., 2018*). These findings indicate that the remodeling capacity of microvascular ECs during obesity is vital not only for the adipose tissue function but also for the

**eLife digest** In the body, thread-like blood vessels called capillaries weave their way through our tissues to deliver oxygen and nutrients to every cell. When a tissue becomes bigger, existing vessels remodel to create new capillaries that can reach far away cells. However, in obesity, this process does not happen the way it should: when fat tissues expand, new blood vessels do not always grow to match. The starved fat cells can start to dysfunction, which causes a range of issues, from inflammation and scarring of the tissues to problems with how the body processes sugar and even diabetes. Yet, it is still unclear why exactly new capillaries fail to form in obesity.

What we know is that a protein called FoxO (short for Forkhead box O) is present in the cells that line the inside of blood vessels, and that it can stop the development of new capillaries. FoxO controls how cells spend their energy, and it can force them to go into a resting state. During obesity, the levels of FoxO actually increase in capillary cells. Therefore, it may be possible that FoxO prevents new blood vessels from growing in the fat tissues of obese individuals.

To find out, Rudnicki et al. created mice that lack the FoxO protein in the cells lining the capillaries, and then fed the animals a high-fat diet. These mutant mice had more blood vessels in their fat tissue, and their fat cells looked healthier. They also stored less fat than normal mice on the same diet, and their blood sugar levels were normal. This was because the FoxO-deprived cells inside capillaries were burning more energy, which they may have obtained by pulling sugar from the blood.

These results show that targeting the cells that line capillaries helps new blood vessels to grow, and that this could mitigate the health problems that arise with obesity, such as high levels of sugar (diabetes) and fat in the blood. However, more work is needed to confirm that the same cellular processes can be targeted to obtain positive health outcomes in humans.

DOI: https://doi.org/10.7554/eLife.39780.002

development of systemic metabolic disturbances. However, surprisingly little is understood about the signaling pathways that limit the angiogenic response of EC in obesity.

Forkhead Box O1 (FoxO1) signaling is essential to the homeostasis of EC and restricts vascular growth (*Wilhelm et al., 2016*). In addition to the control of angiogenesis-related genes (*Potente et al., 2005*; *Paik et al., 2007*; *Milkiewicz et al., 2011*; *Roudier et al., 2013*; *Wilhelm et al., 2016*), FoxO1 is a gatekeeper of EC metabolism; its overexpression reduces the metabolic rate of EC and enforces a state of endothelial quiescence (*Wilhelm et al., 2016*). Thus, this transcription factor is one of the major regulators of angiogenic capacity, since the switch from a quiescent to an angiogenic phenotype requires a coordinated increase in EC metabolic activity to meet the higher demand for energy and biomass production associated with proliferation and migration (*De Bock et al., 2013*; *Schoors et al., 2015*; *Kim et al., 2017*).

Compelling observational evidence indicates that endothelial FoxO1 dysregulation coincides with obesity-associated metabolic disturbances. For instance, FoxO1 protein levels were elevated in capillaries from skeletal muscle of mice fed a high-fat diet (*Nwadozi et al., 2016*) and the activity of endothelial FoxO1 correlated with adipose insulin resistance of obese subjects (*Karki et al., 2015*). Additionally, in vitro conditions that mimic hyperglycemia and insulin resistance increase FoxO1 protein and activity in EC (*Tanaka et al., 2009*; *Nwadozi et al., 2016*). Nevertheless, to our knowledge, the contribution of FoxO1 signaling to vascular remodeling during obesity has not been addressed experimentally. To date, only a few reports have assessed the relevance of endothelial FoxO proteins in diet-induced disorders, but none have examined the influence on adipose tissue. Moreover, those studies employed simultaneous EC-specific depletion of multiple FoxOs (FoxO1, FoxO3, and FoxO4) and transgenic lines in which gene targeting was not exclusive to EC (*Tanaka et al., 2009*; *Tsuchiya et al., 2012*; *Nwadozi et al., 2016*), preventing discrimination of the specific functions of endothelial FoxO1. Notably, it has been shown that in vitro conditions associated with FoxO1 dysregulation can also compromise EC metabolism (*Zhang et al., 2000*; *Du et al., 2003*; *Jais et al., 2016*). Although this suggests that the interplay between endothelial FoxO1 levels and EC metabolic activity may be critically implicated in limiting vascular remodeling in obesity, this concept demands validation.

The converging roles of FoxO1 in the angiogenic phenotype and the metabolism of quiescent EC led us to hypothesize that FoxO1 is a critical nodal point in determining the response of capillary EC to obesity. Consequently, we postulated that targeted endothelial-specific depletion of FoxO1 would provoke capillary growth, preventing obesity-driven adipose tissue dysfunction, and provide a valuable tool to unmask the role of the microvascular endothelium metabolism in the pathophysiology of obesity.

## Results

### Mice with EC-FoxO1 depletion exhibit greater vascular density in visceral adipose tissue

To assess the involvement of EC-FoxO1 in the control of vascular growth in the adipose tissue of adult mice, we utilized a mouse model of EC-selective depletion of FoxO1 expression (referred to 'EC-FoxO1 KD' mice hereafter) through inactivation of the *Foxo1* gene specifically in EC. *Foxo1* floxed (*Foxo1*$^{f/f}$) mice were crossbred with *Pdgfb*-iCreERT2 mice that express tamoxifen-activated Cre recombinase in EC and Cre-mediated recombination of *Foxo1* was induced in adult mice. Littermate mice homozygous for the floxed *Foxo1* allele but not expressing Cre recombinase were used as controls. After tamoxifen injection, *Foxo1* recombination was observed within adipose and skeletal muscle but not within the liver, as endothelial *Pdgfb* expression is undetectable in this organ (*Hellström et al., 1999*). The endothelial cell specificity of the recombinase activity was confirmed in microvascular EC isolated from adipose tissue (*Figure 1A*). Consequently, *Foxo1* transcript level, as measured by qPCR, was decreased by 50% in microvascular EC of EC-FoxO1 KD mice relative to control littermates 8 weeks after the administration of tamoxifen, confirming effective and stable *Foxo1* depletion in these cells (*Figure 1B*). Of note, *Foxo3* mRNA expression was unaltered in microvascular EC (*Figure 1B*), demonstrating a lack of compensation by this FoxO family member in response to the depletion of *Foxo1*. Moreover, consistent with the previously described absence of *Pdgfb*-Cre activity within macrophages (*Claxton et al., 2008*), no significant changes in *Foxo1* mRNA levels were detected in CD16/CD32$^+$ immune cells from white adipose tissue (*Figure 1C*), indicating that Cre-mediated recombination did not occur in these stromal cells. The depletion of

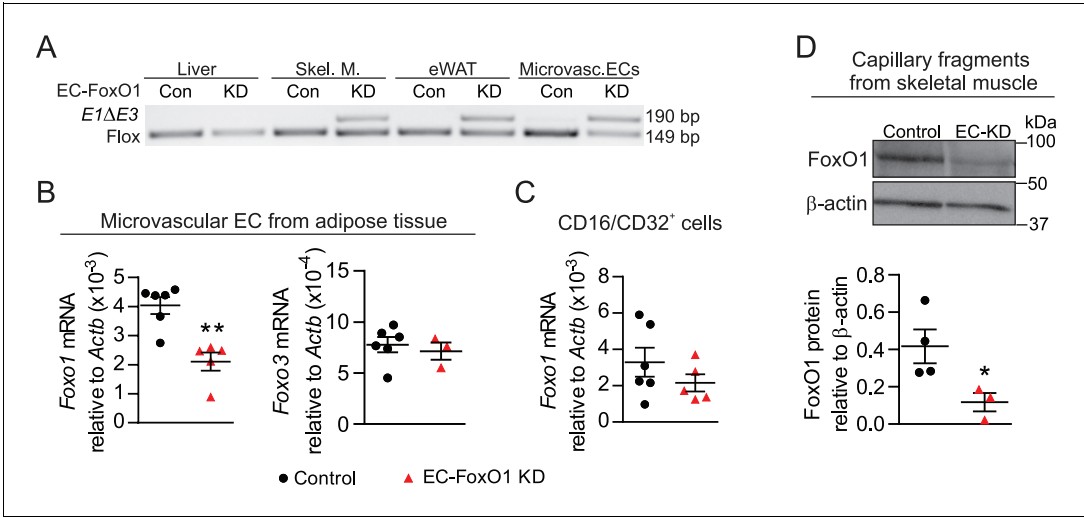

**Figure 1.** Endothelial-specific depletion of *Foxo1* induced in adult male mice effectively reduces FoxO1 levels in skeletal muscle and adipose microvascular beds. (**A**) PCR of genomic DNA from multiple organs of control (Cre⁻; *Foxo1*$^{f/f}$) and EC-FoxO1 KD mice using primers for the floxed and deleted (E1ΔE3) alleles. (**B–C**) Gene expression analysis of microvascular EC and CD16/CD32$^+$ cells isolated from white adipose tissue of Control (n = 6) and EC-FoxO1 KD (n = 3–5) mice. (**D**) Representative Western blot images and quantitative analysis of FoxO1 and β-actin levels in capillary fragments isolated from skeletal muscle (n = 3–4). Results are expressed relative to β-actin levels. Data in all panels are expressed as mean ± SEM; *p < 0.05, **p < 0.01, calculated with two-tailed unpaired *t*-test.
DOI: https://doi.org/10.7554/eLife.39780.003

EC-*Foxo1* in microvascular beds of EC-FoxO1 KD mice also was validated via assessment of FoxO1 protein levels by Western blotting. In agreement with the lower mRNA levels observed in microvascular ECs from adipose tissue, protein levels of FoxO1 were diminished by 70% in capillary fragments from skeletal muscle of EC-FoxO1 KD mice compared to control littermates 6 weeks after the administration of tamoxifen (*Figure 1D*). Together, these results not only imply that successful *Foxo1* depletion was constrained to the endothelial cell compartment, particularly microvascular beds, but also support the use of EC-FoxO1 KD mice as an appropriate model to assess the relevance of endothelial FoxO1 for vascular remodeling during adipose tissue expansion.

EC-FoxO1 KD mice maintained on a normal chow (NC) diet for 16 weeks exhibited no gross abnormalities and similar body weight gain compared to control counterparts (8.32 ± 1.3 *vs.* 7.75 ± 1.17 g, n = 6/group), but significantly increased mRNA levels of the EC marker *Pecam1* in eWAT (*Figure 2A*). When blood vessels were visualized by whole-mount staining with *G. simplicifolia* lectin, it was evident that the vascular density of visceral adipose tissue from EC-FoxO1 KD mice (*Figure 2B–C*) was significantly higher. EC-FoxO1 depletion did not alter the number of vessel branch points (*Figure 2D*). On the other hand, vessels in the adipose of EC-FoxO1 KD mice were significantly enlarged, showing increased vessel diameter, compared to control littermates (*Figure 2E*), which was consistent with the reported influence of EC-*Foxo1* depletion in vascular development in retinas (*Wilhelm et al., 2016*). No difference in the expression of *Pecam1* was detected in other assessed tissues, such as skeletal muscle and liver (*Figure 2A*).

## EC-FoxO1 depletion provokes greater microvascular remodeling under the stimulus of a high-fat diet

To determine whether EC-FoxO1 depletion evokes vascular growth during adipose expansion in response to excess caloric consumption, we challenged mice with a prolonged high-fat diet (HF) and assessed tissue angiogenesis. Gene expression analysis indicated that EC-FoxO1 depletion resulted in higher *Pecam1* mRNA levels in multiple adipose tissue depots: eWAT, subcutaneous and brown adipose tissue (BAT, *Figure 3A*). In line therewith, transcript levels of other EC markers, von Willebrand factor (*Vwf*) and endothelial nitric oxide synthase (*Nos3*) were elevated in the eWAT of HF-fed EC-FoxO1 KD mice (*Figure 3B*). Whole-mount staining of adipose tissue revealed remarkable increases in vascular area and number of vessel branch points in the eWAT of HF-fed EC-FoxO1 KD (*Figure 3C–E*). Consistently, quantitative histological analysis showed that capillary number per adipocyte (capillary to adipocyte ratio) was significantly higher in eWAT of HF-fed EC-FoxO1 KD mice, further validating the greater microvascular content in eWAT of these mice compared to HF-fed control counterparts (*Figure 3F–G*). Furthermore, EC-FoxO1 depletion led to significant capillary enlargement in eWAT (*Figure 3H–I*). Of note, the increase of vessel diameter in HF-fed EC-FoxO1 KD mice was greater than observed in NC-fed EC-FoxO1 KD mice (1.8 *vs.* 1.3-fold increase),

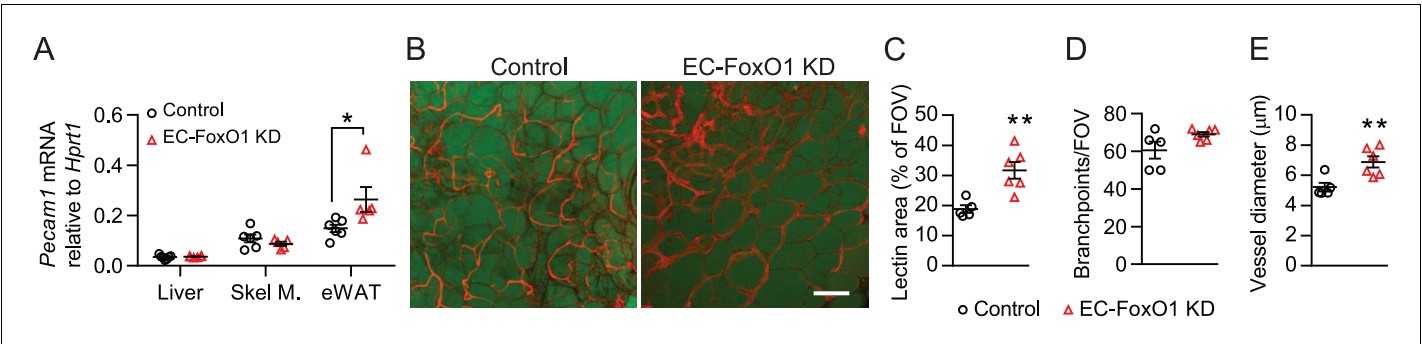

**Figure 2.** Greater vascular density in visceral adipose tissue of normal chow-fed EC-FoxO1 KD mice. (A) *Pecam1* mRNA levels in various tissues of Control and EC-FoxO1 KD mice after 16 weeks of normal chow (NC) diet (Control n = 6, EC-FoxO1 KD n = 5). (B) Representative confocal images of adipose tissue whole-mount staining with BODIPY 493/503 (green) and *G. simplicifolia* lectin (red) (×20 magnification; scale bar = 100 μm). (C–E) Lectin area (C), capillary branch density (D) and microvessel diameters (E) were quantified from confocal images (Control n = 5, EC-FoxO1 KD n = 6). Data in all panels are expressed as mean ± SEM; *p < 0.05, **p < 0.01, calculated with two-tailed unpaired *t*-test.

DOI: https://doi.org/10.7554/eLife.39780.004

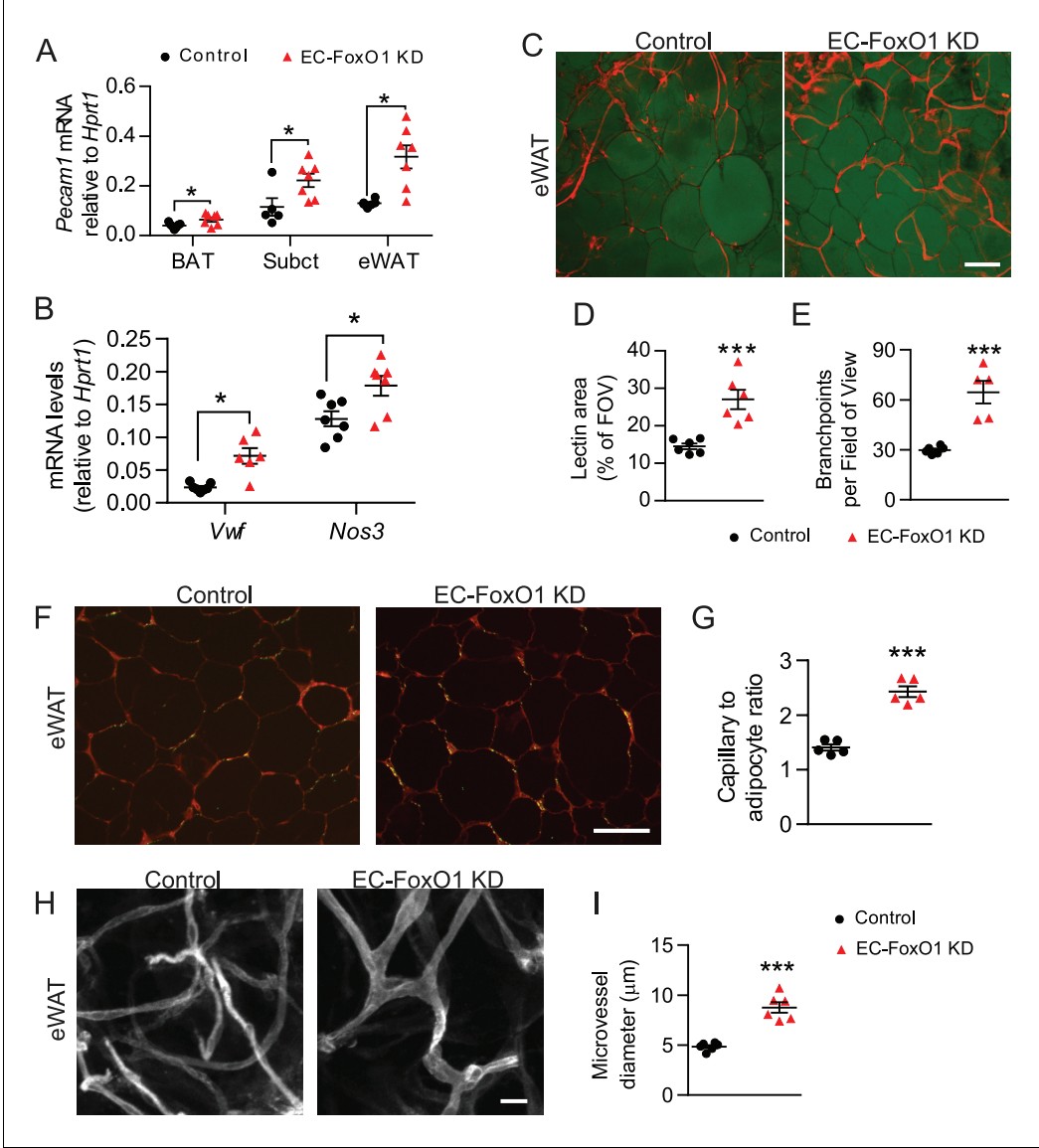

**Figure 3.** EC-*Foxo1* depletion strongly induces vascular growth within adipose tissue in response to HF diet. (**A**) *Pecam1* mRNA levels in different adipose tissue depots of Control and EC-FoxO1 KD mice after 16 weeks of high-fat (HF) diet (Control n = 5–7, EC-FoxO1 KD n = 7). (**B**) Gene expression analysis of eWAT of HF-fed Control and EC-FoxO1 KD mice (Control n = 7, EC-FoxO1 KD n = 6–7). (**C**) Representative confocal images of adipose tissue whole-mount staining with BODIPY 493/503 (green) and *G. simplicifolia* lectin (red) (C - scale bar = 100 μm). (**D,E and I**). Lectin area (**D**) and capillary branch density were quantified from these images (Control, n = 6; EC-FoxO1 KD, n = 5 or 6). (**F–G**) *G. simplicifolia* lectin (green) and Wheat germ agglutinin (red) staining of paraffin-sectioned adipose tissue (F - scale bar = 100 μm) was used to assess capillary to adipocyte ratio (**G**). (**H**) Representative confocal images of adipose tissue whole-mount staining with Isolectin alone (greyscale; scale bar = 20 μm). (**I**) Microvessel diameters were quantified from confocal images (Control, n = 6; EC-FoxO1 KD, n = 6). Data in all panels are expressed as mean ± SEM; *p < 0.05, ***p < 0.001, calculated with two-tailed unpaired *t*-test.
DOI: https://doi.org/10.7554/eLife.39780.005

suggesting that the enlargement of capillaries promoted by EC-FoxO1 depletion is exacerbated by HF feeding.

Subsequent gene expression analysis showed that EC-FoxO1 depletion did not change transcript levels of *Pecam1* in the liver (corresponding with the lack of Cre recombination in this organ) but did upregulate its expression in skeletal muscle, suggesting that under the stimulus of HF diet, EC-

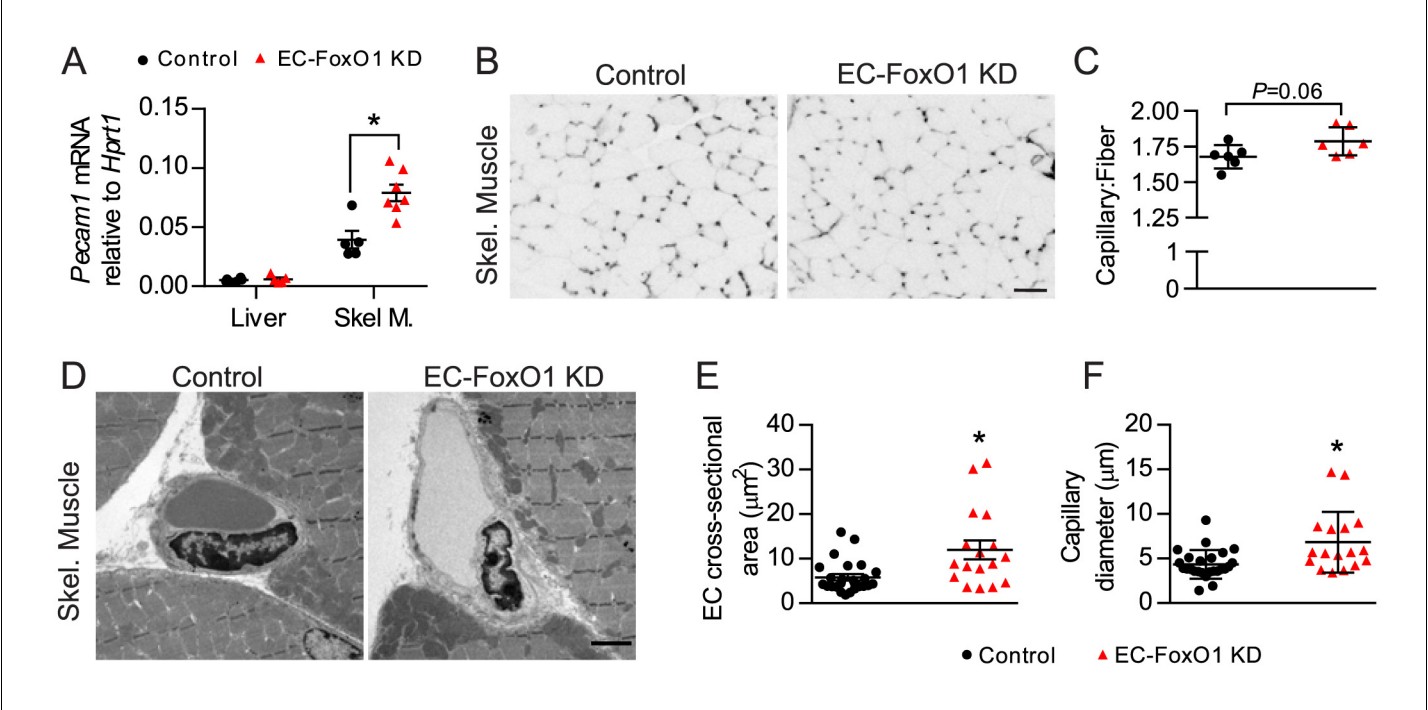

**Figure 4.** EC-*Foxo1* depletion also favors microvascular expansion in skeletal muscle under HF diet feeding. (**A**) *Pecam1* mRNA levels in liver and skeletal of HF-fed Control (n = 5–6) and EC-FoxO1 KD (n = 6–7) mice. (**B**) Images of EDL muscle stained with Isolectin-FITC to identify capillaries (scale bar = 50 µm). (**C**) Capillary to fiber (C:F) ratios were calculated from 3 to 4 independent fields of view per mouse (Control n = 6, EC-FoxO1 KD n = 6). (**D**) Representative EM images of capillaries within skeletal muscle from HF-fed Control and EC-FoxO1 KD mice (×6.5k magnification; scale bar = 2 µm). (**E–F**) EC cross-sectional area (**E**) and capillary luminal diameter were quantified from EM images from n = 4 mice per group, with individual capillary measurements shown (**F**). Data in all panels are expressed as mean ± SEM; *p < 0.05, calculated with two-tailed unpaired *t*-test.
DOI: https://doi.org/10.7554/eLife.39780.006

FoxO1 depletion also induces microvascular remodeling in this tissue (*Figure 4A*). Skeletal muscle of HF-fed EC-FoxO1 KD mice displayed a trend towards higher capillary:fiber ratio (p = 0.06) compared to control mice (*Figure 4B–C*). Transmission electron microscopy revealed increased capillary endothelial cross-sectional area and capillary lumen diameters in skeletal muscle of EC-FoxO1 KD mice, demonstrating a modest expansion of the size of individual capillaries (*Figure 4D–F*). Taken together, these data indicate that EC-FoxO1 depletion results in remarkable vascular growth in response to HF diet, which is particularly pronounced within visceral adipose tissue.

## EC-FoxO1 KD mice exhibit a healthier adipose tissue expansion in response to HF diet

The vasculature is critical for maintenance of adipose tissue homeostasis during obesity-driven adipocyte enlargement. Thus, we inferred that the increased vascular density observed with EC-FoxO1 depletion may hinder adipose tissue expansion and dysfunction induced by high-fat diet. Although HF-fed EC-FoxO1 KD mice showed only a trend towards reduced body weight gain (p = 0.06), these mice displayed less fat accumulation, showing lower trunk fat content, smaller retroperitoneal (rWAT) and subcutaneous fat pads compared to control mice (*Figure 5A–C* and *Table 1*). The phenotype was not explained by changes in food consumption (*Figure 5—figure supplement 1A*). HF-fed EC-FoxO1 KD mice also displayed lower fed levels of serum triglycerides and glycerol, and less hepatic lipid accumulation (*Figure 5—figure supplement 1B–D*), suggesting an improvement in the capacity to handle dietary nutrient excess in these mice. Moreover, histological analysis revealed that increased vascular growth in adipose tissue of HF-fed EC-FoxO1 KD mice was associated with smaller-sized and generally spherical adipocytes, whereas adipocytes from HF-fed control mice were large with irregular polygonal shapes (*Figure 5D–E*), which was previously related to cellular stress

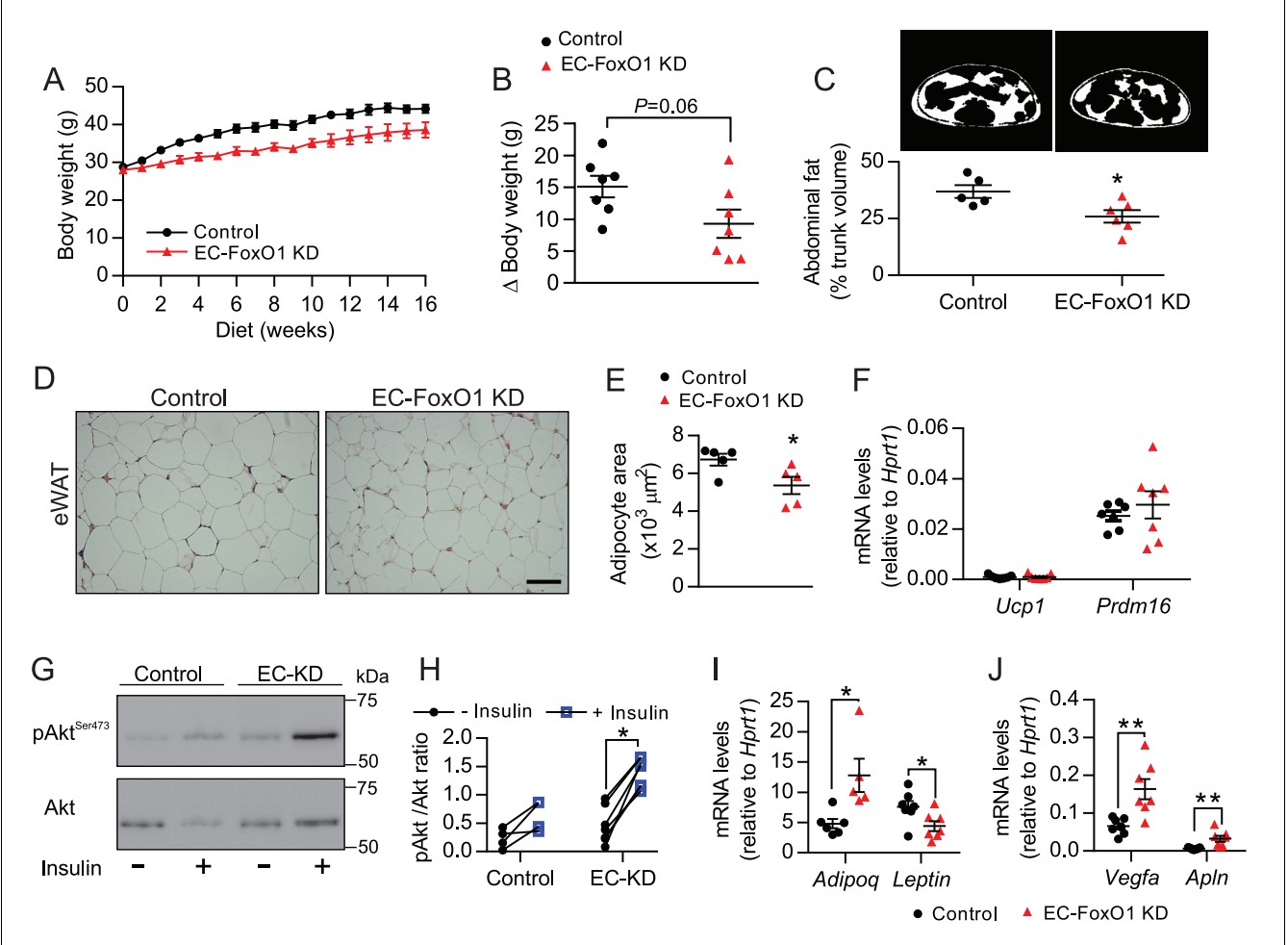

**Figure 5.** EC-FoxO1 KD mice exhibit a healthier adipose tissue expansion in response to HF diet. (**A**) Body weights during 16 weeks of HF feeding. (**B**) Summarized weight gain over the course of 0–14 weeks (Control n = 7, EC-FoxO1 KD n = 7). (**C**) Abdominal transverse micro-CT images of HF-fed Control (n = 5) and EC-FoxO1 KD (n = 6) mice (upper panel). Fat content (shown in white) was calculated as % of total trunk volume. (**D**) Representative hematoxylin and eosin-stained images of adipose tissue from the epididymal fat pad (scale bar = 100 µm). (**E**) Mean adipocyte cross-sectional area (Control n = 5 EC-FoxO1 KD n = 5). (**F**) mRNA for browning markers *Ucp1* and *Prdm16* relative to *Hprt1* (Control n = 7, EC-FoxO1 KD n = 7). (**G–H**) Representative Western blot images (**G**) and quantitative analysis (**H**) of pSer473-Akt and total Akt levels in eWAT after ex vivo incubation in the absence or presence of insulin. Results are expressed relative to total Akt levels (Control n = 4, EC-FoxO1 KD n = 7). (**I–J**) mRNA for adipokines (I, *Adipoq* and *Leptin*) and angiogenic markers (J, *Vegfa* and *Apln*) in eWAT relative to *Hprt1* (Control n = 6–7, EC-FoxO1 KD n = 5–7). Data in all panels are expressed as mean ± SEM; *p < 0.05, **p < 0.01, calculated with two-tailed unpaired *t*-test.

DOI: https://doi.org/10.7554/eLife.39780.007

The following figure supplements are available for figure 5:

**Figure supplement 1.** Lower circulating and liver triglycerides in HF-fed EC-FoxO1 KD mice.
DOI: https://doi.org/10.7554/eLife.39780.008

**Figure supplement 2.** EC-*Foxo1* depletion has no impact on adipose mitochondrial content and respiration, or sensitivity to isoproterenol.
DOI: https://doi.org/10.7554/eLife.39780.009

(***Giordano et al., 2013***). Of note, adipocytes from EC-FoxO1 KD mice retained a unilocular structure (***Figure 5D***) rather than the hallmark multilocular morphology of brown fat. Furthermore, no change in the mRNA levels of browning markers *Ucp1* and *Prdm16* (***Figure 5F***) was detected with EC-FoxO1 depletion. Correspondingly, we did not observe any difference in mitochondrial protein content of eWAT nor in ADP-stimulated respiration through either Complex I (pyruvate/malate, glutamate) or Complex II (succinate) (***Figure 5—figure supplement 2A–C***). Isoproterenol-stimulated phosphorylation of the hormone-sensitive lipase (HSL) was unaffected (***Figure 5—figure supplement 2D–E***), indicating that EC-FoxO1 depletion did not impact the adipose tissue sensitivity to lipolytic

**Table 1.** Tissue weights of Control and EC-FoxO1 KD mice after 16 weeks of HF diet

|  | Control | EC-FoxO1 KD |
| --- | --- | --- |
| Body weight (g) | 44 ± 1.1 | 38 ± 2.2 |
| eWAT (g) | 1.8 ± 0.2 | 1.6 ± 0.6 |
| rWAT (mg) | 881.7 ± 86.1 | 579.8 ± 98.7[*] |
| Subcutaneous adipose (g) | 2.1 ± 0.3 | 1.3 ± 0.2[*] |
| BAT (mg) | 189.2 ± 21.2 | 144.3 ± 14.8 |
| Liver (g) | 1.5 ± 0.1 | 1.2 ± 0.1[*] |
| Heart (mg) | 135 ± 4.2 | 126.6 ± 3.4 |
| Gastrocnemius (mg) | 130.9 ± 4.3 | 115.5 ± 4.4[*] |
| Soleus (mg) | 8.3 ± 0.5 | 8.0 ± 0.5 |
| Tibialis anterior (mg) | 44.5 ± 1.5 | 42.6 ± 1.4 |

eWAT: epididymal adipose tissue; rWAT: retroperitoneal adipose tissue; BAT: brown adipose tissue

Data are expressed as mean ± SEM, n = 7 per group

Significance was established using unpaired *t*-test

[*]*P* < 0.05 *vs* Control group

DOI: https://doi.org/10.7554/eLife.39780.010

stimuli. In contrast, and consistent with an improved function, eWAT from HF-fed EC-FoxO1 KD mice displayed enhanced Akt phosphorylation in response to insulin (*Figure 5G–H*), which was accompanied by higher *Adiponectin* mRNA levels and concomitant lower *Leptin* expression (*Figure 5I*). Collectively, these findings demonstrate that depletion of EC-FoxO1 signaling exerts a protective effect against obesity-induced metabolic remodeling of adipose tissue without promoting a browning phenotype. Notably, the transcripts levels of *Vegfa* and *Apelin* were also higher in eWAT from HF-fed EC-FoxO1 KD mice (*Figure 5J*), providing evidence that the improvements in adipose phenotype include a more pro-angiogenic adipose tissue microenvironment.

## EC-FoxO1 depletion provokes a metabolic shift favoring glucose utilization in HF-fed mice

To better understand the metabolic consequences of EC-FoxO1 depletion, whole-body metabolic functions were monitored for 48 hr in a 2nd cohort of HF-fed mice. Surprisingly, EC-FoxO1 KD mice exhibited reduced $VO_2$ and increased RER during the dark cycle (*Figure 6A,C*) with equivalent $CO_2$ production and activity levels compared to control mice (*Figure 6B,D*). These data unexpectedly indicated that EC-FoxO1 KD mice increased oxidation of carbohydrate relative to fatty acid as an energy substrate, suggesting that EC-FoxO1 depletion shifted whole-body energy homeostasis towards glucose oxidation. Consistent with these findings, HF-fed EC-FoxO1 KD mice displayed more rapid glucose clearance from the blood during glucose tolerance tests (*Figure 6E–F*). However, higher glucose tolerance was not associated with altered whole-body insulin sensitivity, based on insulin tolerance tests (ITT) (*Figure 6G–H*) or insulin-mediated Akt phosphorylation in the skeletal muscle (*Figure 6—figure supplement 1*). Despite the effects on glucose metabolism observed in HF-fed EC-FoxO1 KD mice, no change in whole-body glucose metabolism was detected in NC-fed EC-FoxO1 KD mice compared to control counterparts (*Figure 6—figure supplements 1* and *2*). Interestingly, fasting glycemia was significantly lower (*Figure 6I*) whereas serum lactate levels were elevated in HF-fed EC-FoxO1 KD mice compared to their littermates (10.9 ± 0.39 *vs.* 9.4 ± 0.55 mmol/L, respectively, p = 0.04, n = 7/group). These findings imply that altered whole-body glucose metabolism of EC-FoxO1 KD mice on a HF may be due to higher glucose turnover, leading us to postulate that increased glycolytic rates at the tissue level contribute to the metabolic phenotype of HF- EC-FoxO1 KD mice. To address this question, we first assessed the expression of main glycolytic genes, including the constitutive glucose transporter GLUT1 (*Slc2a1*), the rate-limiting enzymes hexokinase 2 (*Hk2*) and phosphofructokinase (*Pfkm*) and phosphofructokinase-2/fructose-2,6-bisphosphatase-3 (*Pfkfb3*). As anticipated, the mRNA levels of most glycolytic genes, with the exception of *Pfkfb3*, were upregulated in the eWAT from HF-fed EC-FoxO1 KD mice compared to control

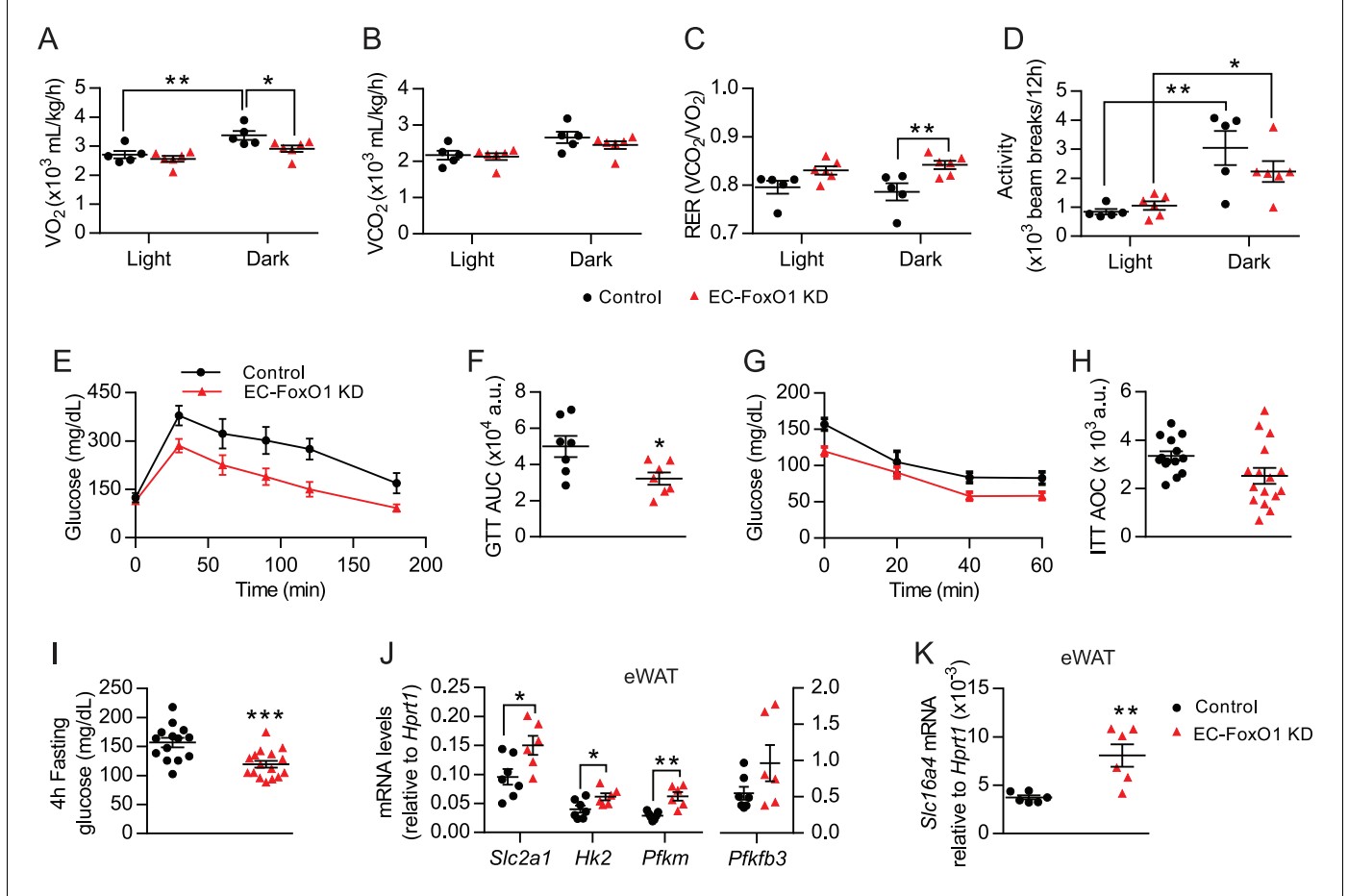

**Figure 6.** EC-*Foxo1* depletion improves glucose homeostasis in HF-fed mice. (A–D) $O_2$ consumption (A), $CO_2$ production (B), Respiratory exchange ratio - RER (C) and daily activity (D) were measured during indirect calorimetry tests using a comprehensive laboratory animal monitoring system (CLAMS, Control n = 5, EC-FoxO1 KD n = 6). (E) Glucose tolerance of HF-fed Control and EC-FoxO1 KD mice was examined by intraperitoneal glucose tolerance test after 15 weeks of HF diet and 16 hr fasting. (F) Area under the curve (AUC, Control n = 7, EC-FoxO1 KD n = 7). (G) Insulin sensitivity of HF-fed Control and EC-FoxO1 KD mice was assessed by intraperitoneal insulin tolerance test after 14 weeks of HF diet and 4 hr fasting. (H) Area over the curve (AOC, Control n = 14, EC-FoxO1 KD n = 16). (I) Plasma glucose levels of HF-fed Control (n = 14) and EC-FoxO1 KD (n = 16) mice after 4 hr fasting. (J–K) eWAT gene expression analysis by qPCR (Control n = 6–7, EC-FoxO1 KD n = 6). Data in all panels are expressed as mean ± SEM; *p < 0.05, **p < 0.01, ***p < 0.001, calculated with two-tailed unpaired *t*-test (F, I, J, K) or post hoc Bonferroni-corrected *t*-tests when a statistical significance was detected by two-way ANOVA model (A,C,D).

DOI: https://doi.org/10.7554/eLife.39780.011

The following figure supplements are available for figure 6:

**Figure supplement 1.** EC-*Foxo1* depletion has no effect on muscle insulin sensitivity.
DOI: https://doi.org/10.7554/eLife.39780.012
**Figure supplement 2.** EC-*Foxo1* depletion does not affect glucose homeostasis in NC-fed mice.
DOI: https://doi.org/10.7554/eLife.39780.013

mice, (*Figure 6J*). Furthermore, mRNA levels of the lactate transporter, monocarboxylate transporter 5, *Slc16a4*, were also increased in eWAT of HF-fed EC-FoxO1 KD mice (*Figure 6K*), consistent with an increased glycolytic flux of glucose to lactate in the adipose tissue of these mice.

## EC drive the changes in glucose metabolism

EC rely dominantly on glycolysis to support angiogenesis (*De Bock et al., 2013*) and a previous study reported that FoxO1 overexpression represses EC metabolism (*Wilhelm et al., 2016*). Therefore, we hypothesized that the changes in glucose utilization were due, at least in part, to increased metabolic activity of EC resulting from EC-FoxO1 depletion. To explore this possibility, we isolated

the EC fraction from white adipose tissue depots of mice fed a HF diet for 7 weeks and first assessed gene expression of main glycolytic pathway genes. Consistent with the findings observed with whole adipose tissue, increased mRNA levels of glycolytic genes *Slc2a1*, *Pfkm* and *Pfkfb3* were detected in the EC fraction from adipose tissue of HF-fed EC-FoxO1 KD mice (*Figure 7A–B*). We also tested whether the elevated gene expression of glycolytic markers in EC from EC-FoxO1 KD mice would correspond with greater glycolytic capacity, as assessed by cellular glucose uptake and changes in glucose consumption and the accumulation of lactate. In agreement with higher transcript levels of *Slc2a1*, EC freshly isolated from adipose tissue of HF-fed EC-FoxO1 KD mice displayed increased glucose uptake than EC from floxed controls (*Figure 7C*). Moreover, rates of glucose consumption and lactate production were higher in EC with FoxO1 depletion compared to control cells (*Figure 7D–E*). Additionally, we observed elevated *Mki67* mRNA in the EC fraction from EC-FoxO1 KD mice, indicating that an enhanced proliferative state coincides with the glycolytic activity of these EC (*Figure 7F*).

To corroborate that dysregulation of FoxO1 signaling is directly involved in disruption of glycolytic processes, we cultured skeletal muscle EC in low (5 mmol/L) and high glucose (25 mmol/L) conditions, as previous in vitro studies have shown that hyperglycemia can both increase FoxO1 activity (*Tanaka et al., 2009*) and stall EC metabolism (*Du et al., 2000*; *Zhang et al., 2000*; *Du et al., 2003*). As expected, high-glucose significantly increased FoxO1 protein levels in cultured EC (*Figure 8A–B*) and provoked changes in established FoxO1 target genes *Cdkn1b* (p27) and *Ccnd1* (cyclin D1) (*Figure 8—figure supplement 1A–B*) that are involved in cell proliferation. Consistent with increased FoxO1 levels and activity, high-glucose conditions also lowered the mRNA levels of glycolytic pathway components *Slc2a1*, *Hk2*, and *Pfkfb3* (*Figure 8C–E*). Importantly, pharmacological inhibition of FoxO1 significantly reversed the high-glucose-induced reduction of each of these genes (*Figure 8C–E*), which correlated with elevated protein levels of HK2 and PFKFB3 (*Figure 8F–I*). Accordingly, treatment with FoxO1 inhibitor AS1842856 increased cellular glucose uptake and

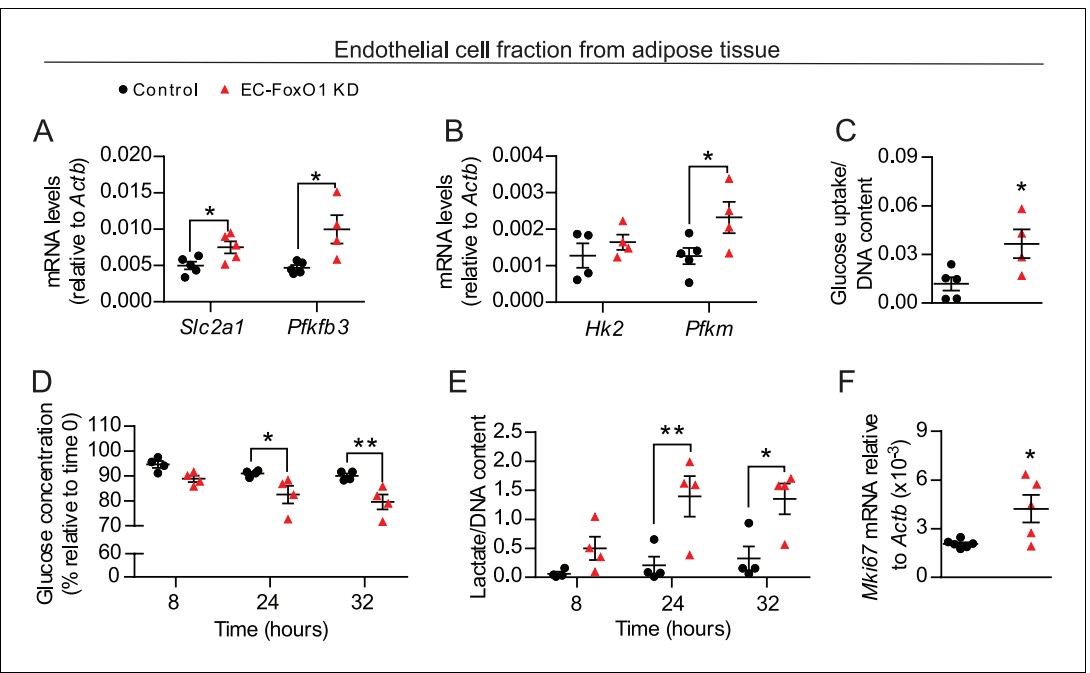

**Figure 7.** FoxO1 is a critical regulator of glucose metabolism in EC. (**A, B, F**) Gene expression analysis of EC fraction from adipose tissue from Control (n = 3–6) and EC-FoxO1 KD (n = 4–5) mice fed a HF diet for 7 weeks. (**C–E**) Increased glucose uptake (**C**) glucose consumption (**D**) and lactate production (**E**) in EC fraction from HF-fed EC-FoxO1 KD (n = 4) mice compared to Control (n = 4–5). Data in all panels are expressed as mean ± SEM; *p < 0.05, **p < 0.01, calculated with two-tailed unpaired *t*-test (**A,B,C,F**) or post hoc Bonferroni-corrected *t*-tests when a statistical significance was detected by two-way ANOVA model (**D,E**).
DOI: https://doi.org/10.7554/eLife.39780.014

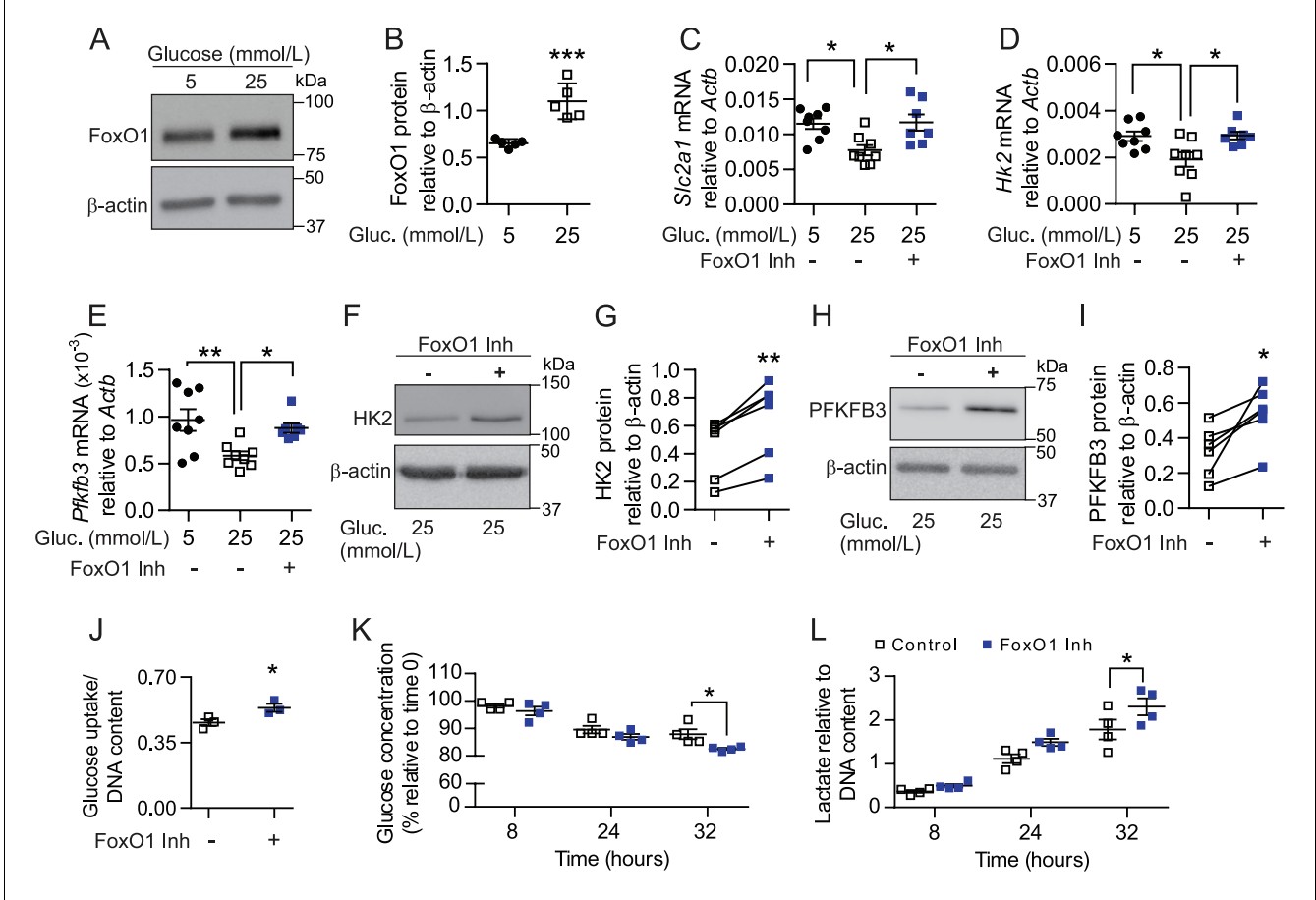

**Figure 8.** Pharmacological inhibition of FoxO1 in SMEC reproduces in vitro the endothelial phenotype observed with EC-*Foxo1* depletion. (**A–B**) Representative Western blot images (**A**) and quantitative analysis (**B**) of FoxO1 and β-actin levels in primary EC from skeletal muscle cultivated under low (5 mmol/L) or high (25 mmol/L) glucose conditions for 48 hr (n = 5). Results are expressed relative to β-actin levels. (**C–E**) Transcript analysis by qPCR of microvascular EC from skeletal muscle cultivated under low (5 mmol/L, n = 8) or high (25 mmol/L, n = 8) glucose conditions for 48 hr in the presence or absence of the FoxO1 inhibitor (1 μmol/L AS1842856, n = 7) in the last 18 hr. (**F–I**) Representative Western blot images and quantitative analysis of hexokinase II (HK2, (**F,G**), PFKFB3 (**H, I**) and β-actin levels in primary EC from skeletal muscle cultivated under high glucose (25 mmol/L) conditions and treated with 1 μmol/L AS1842856 for 24 hr (n = 6). Results are expressed relative to β-actin levels. (**J**) Glucose uptake after 18 hr treatment with 1 μmol/L AS1842856 of microvascular EC from skeletal muscle cultivated under high glucose conditions (n = 3). (**K–L**) Glucose consumption (**K**) and Lactate production (**L**) were assessed in SMEC in the absence or presence of 1 μmol/L AS1842856. Cells were pretreated with 1 μmol/L AS1842856 for 24 hr (n = 4). Data in all panels are expressed as mean ± SEM; *p < 0.05, **p < 0.01, ***p < 0.001, calculated with two-tailed unpaired *t*-test (**B,J**), post hoc Bonferroni-corrected *t*-tests when a statistical significance was detected by two-way ANOVA model (**C–E, K–L**) or two-tailed paired *t*-test (**G,I**).

DOI: https://doi.org/10.7554/eLife.39780.015

The following figure supplement is available for figure 8:

**Figure supplement 1.** High-glucose conditions induce the expression of FoxO1 target genes.

DOI: https://doi.org/10.7554/eLife.39780.016

consumption in microvascular ECs, which corresponded with higher extracellular lactate levels (*Figure 8J–L*). Collectively, these findings indicate that lower FoxO1 levels and activity increase glycolytic and proliferative activities of EC. This induces a profound increase in glucose consumption by these cells, which consequently leads to higher glucose utilization at the tissue level, ultimately impacting whole-body glucose homeostasis.

## EC-FoxO1 depletion is sufficient to alter EC response to HF diet

FoxO1 and FoxO3 can demonstrate overlapping functions in EC (*Potente et al., 2005*) and previous studies reported that double depletion of *Foxo1* and *Foxo3* can either demonstrate similar effects

as FoxO1 deficiency or have significant additive effects (*Zhang et al., 2012*; *Haeusler et al., 2014*). This led us to investigate whether double depletion of endothelial *Foxo1* and *Foxo3* would result in a greater angiogenic response in HF-fed mice. HF-fed EC-FoxO1,3 KD mice (generated using the tamoxifen-inducible, endothelial-specific Cre driver: *Pdgfb*-CreERT2) presented lower levels of fasting glucose (*Figure 9A–C*) and reduced adiposity, as evidenced by lighter subcutaneous and rWAT depots, compared to their littermate controls (*Supplementary file 1*). Similar to what was observed in EC-FoxO1 KD mice on a HF diet, increased expression of *Pecam1* was detected in eWAT and skeletal muscle of HF-fed EC-FoxO1,3 KD mice (*Figure 9D*), indicating greater EC content in these tissues. Additionally, increased expression of glycolytic markers was observed in eWAT and EC fraction from adipose tissue of HF-fed EC-FoxO1,3 KD mice (*Figure 9E–F*). Consistent with these findings, increased glucose uptake (*Figure 9G*) and lactate production (*Figure 9H*) were also detected in the EC fraction from HF-fed EC-FoxO1,3 KD mice. Notably, these measurements demonstrated that combined depletion of EC-*Foxo1* and *Foxo3* elicits a similar, but not additive effect, when compared to EC-*Foxo1* alone, indicating that FoxO1 is the dominant regulator of the EC response to HF diet.

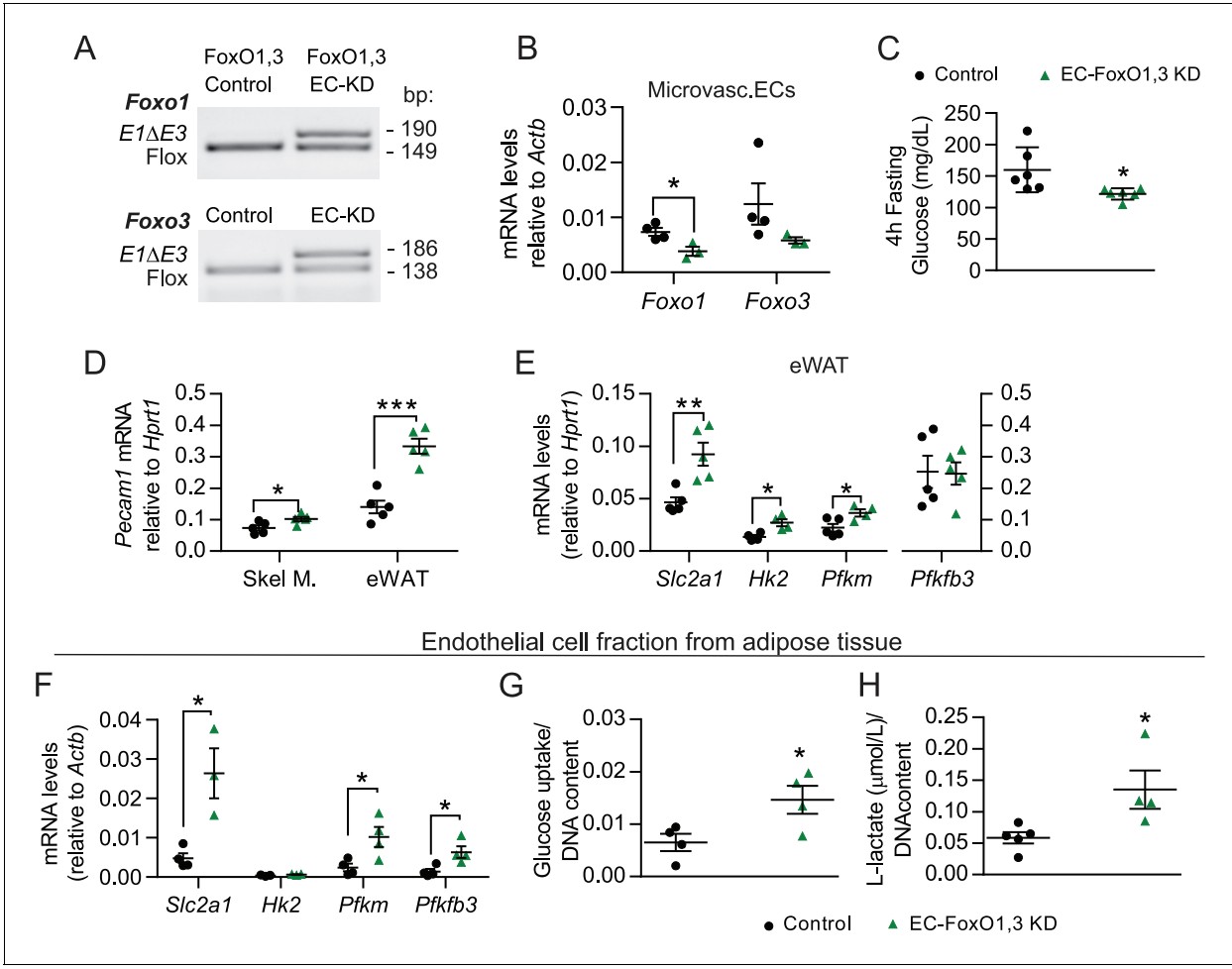

**Figure 9.** EC-*Foxo1,3* depletion increases vascular growth and upregulates endothelial glycolytic processes comparable to EC-*Foxo1* depletion. (**A**) PCR of genomic DNA from eWAT of Control (*Foxo1,3^{f/f}*) and EC-FoxO1,3 KD mice using primers for the floxed and deleted alleles for each gene. (**B**) *Foxo1* and *Foxo3* mRNA levels in microvascular EC from white adipose tissue (Control n = 4, EC-FoxO1,3 KD n = 3). (**C**) Glucose levels after 14 weeks of HF feeding and 4 hr fasting (Control n = 6, EC-FoxO1,3 KD n = 6). (**D**) Endothelial cell marker *Pecam1* mRNA level in skeletal muscle and eWAT of HF-fed Control and EC-FoxO1,3 KD mice (Control n = 5, EC-FoxO1,3 KD n = 5). (**E–F**) Gene expression analysis by qPCR of eWAT (**E**) and EC fraction from adipose tissue (**F**) of HF-fed Control (n = 3–5) and EC-FoxO1,3 KD (n = 3–5) mice. (**G–H**) Increased glucose uptake (**G**) and lactate production (**H**) in EC fraction from adipose tissue of EC-FoxO1,3 KD mice (n = 4) compared to Control counterparts (n = 4–5). Data in all panels are expressed as mean ± SEM; *p < 0.05, **p < 0.01, ***p < 0.001, calculated with two-tailed unpaired *t*-test.

DOI: https://doi.org/10.7554/eLife.39780.017

## Discussion

Herein, we provide evidence that endothelial FoxO1 is critical to the development of metabolic disorders in obesity through the converging actions of controlling metabolic activity and angiogenic fate of the endothelium. Our data underscore that the manipulation of endothelial FoxO1 levels profoundly modifies the endothelial phenotype under an obesogenic diet, with lower levels of EC-FoxO1 evoking increased endothelial metabolism and capillary growth, most robustly detected within visceral adipose. These effects were sufficient not only to prevent the detrimental obesity-driven alterations in visceral adipose tissue but also to elicit increased glucose clearance leading to higher glucose tolerance in HF-fed mice (*Figure 10*). Broadly, these findings provide support for the emerging concept that intrinsic metabolic properties of EC actively influence whole-body energy balance.

A marked increase in vascular density in the adipose tissue was the bona fide phenotypic consequence of EC-FoxO1 depletion, which was strikingly evident under the stress of obesity-related tissue expansion. This demonstrates that endothelial FoxO1 is a prime regulator of adipose tissue microvascular remodeling in adult mice, underlining our hypothesis that FoxO1 levels are directly implicated in limiting the angiogenic response of ECs in obesity. Despite the enlargement of capillaries that was observed in EC-FoxO1 KD mice, these mice displayed a healthier adipose phenotype

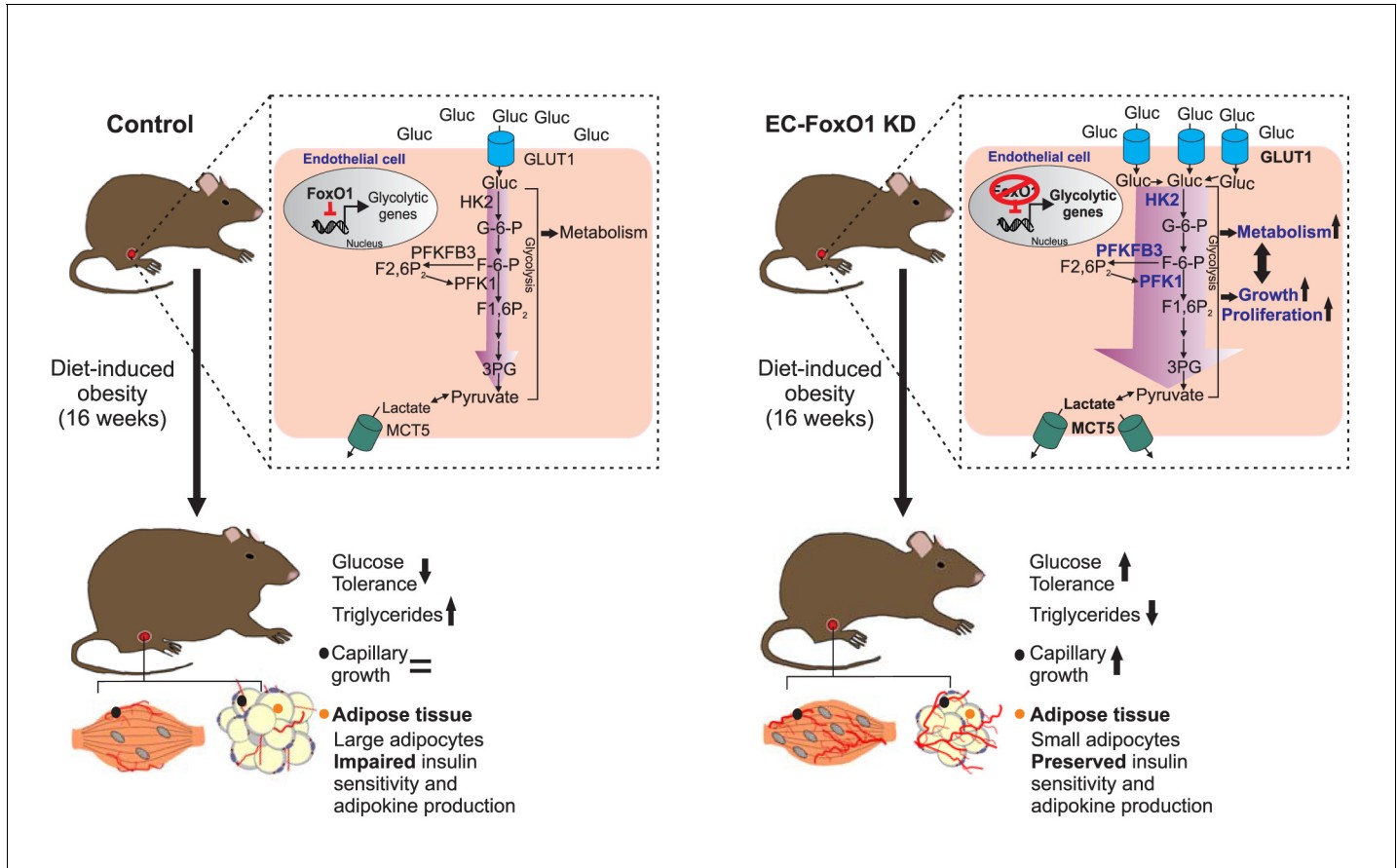

**Figure 10.** Schematic depicting the influence of FoxO1 in controlling the response of ECs to a HF diet. In wild-type mice, FoxO1 represses glycolysis, which prevents endothelial cell growth. This results in impaired angiogenesis during adipose tissue expansion as well as dysfunction of the adipose tissue, which consequently leads to decreased glucose tolerance and increased serum and intra-tissue levels of triglycerides. Conversely, when FoxO1 is depleted in endothelial cells, up-regulation of glycolytic genes accelerates glycolysis, which supports increased cellular metabolism, growth and proliferation. This, in turn, increases the nutrient demand of endothelial cells, resulting in higher uptake and consumption of glucose and an increased production of lactate. The accelerated endothelial cell growth ultimately preserves adipose tissue functions and promotes improved systemic glucose tolerance and lipid metabolism.

DOI: https://doi.org/10.7554/eLife.39780.018

that lacked the metabolic dysfunctions typically caused by obesity. This finding is in line with previous evidence suggesting that enhanced EC-FoxO1 activity is associated with reduced adipocyte insulin sensitivity in the adipose tissue of obese individuals (*Karki et al., 2015*). Therefore, our findings indicate not only that EC-FoxO1 depletion beneficially increases adipose vascular density but also emphasize the intimate interplay of EC and adipocytes and the crucial role of angiogenesis in the maintenance of adipose tissue functions (*Corvera and Gealekman, 2014*; *Crewe et al., 2017*).

In contrast, EC-FoxO1 depletion induced relatively modest expansion of skeletal muscle vasculature. Our findings suggest that the tissue-restricted pattern of FoxO1-driven vascular growth is highly dependent on the co-presence of angiogenic factors within the local environment, which is impacted by nutritional status. Besides the higher levels of angiogenic mediators detected in adipose tissue of EC-FoxO1 KD mice, another argument favoring this hypothesis is provided by our observation that EC-FoxO1 depletion was associated with increased EC content within skeletal muscle only under HF feeding. It is noteworthy that previous reports showed that HF-feeding increases VEGF-A protein within the skeletal muscle (*Silvennoinen et al., 2013*; *Nwadozi et al., 2016*) and FoxO1 levels within skeletal muscle capillaries (*Nwadozi et al., 2016*). In this context, our findings also support the concept that the impaired vascular growth reported with sustained HF diet results from the repressive action of EC-FoxO1 (*Milkiewicz et al., 2011*; *Roudier et al., 2013*) rather than the lack of a pro-angiogenic stimulus.

Interestingly, EC from HF-fed EC-FoxO1 KD mice demonstrated markedly enhanced glycolytic activity, based on the increased expression of glycolytic markers and concomitant increase in glucose uptake, glucose consumption and lactate production. Although the regulation of endothelial metabolism by *Foxo1* overexpression was reported in cultured EC (*Wilhelm et al., 2016*), our data provide novel evidence of the impact of lower FoxO1 levels on endothelial metabolism and its consequences to whole-body homeostasis. It has become recently clear that endothelial metabolic activation represents an important feature of excessive angiogenesis, and that its repression holds therapeutic promise particularly within the tumor microenvironment (*Schoors et al., 2014*; *Cantelmo et al., 2016*). Our study drives this concept from the opposite angle by highlighting the potential for induction of endothelial metabolic activity as an approach to overcome the impairments in adaptive capillary growth that prevail in obese individuals. In fact, our data strongly indicate that the metabolic endothelial adaptation seen in response to FoxO1 depletion results in beneficial expansion of microvascular EC content and prevents obesity-related disorders. The lack of additional influence of combined depletion of EC *Foxo1* and 3 with respect to expression of *Pecam1* and glycolytic pathway genes and glucose uptake reinforces FoxO1 as the dominant regulator of EC metabolic homeostasis. More importantly, these findings indicate FoxO1 as a central target for the manipulation of capillary EC response to obesity-induced conditions.

The finding that increased endothelial glycolysis induced by EC-FoxO1 depletion significantly impacts whole-body energy homeostasis is intriguing. Improvements in whole-body energy homeostasis subsequent to microvascular expansion have been observed previously (*Sun et al., 2012*; *Nwadozi et al., 2016*; *Robciuc et al., 2016*; *An et al., 2017*). Nonetheless, these effects had been ascribed to the improved passive exchange of oxygen, nutrients, and hormones to parenchymal tissues due to the increased capillary EC surface area. Our findings add another dimension to the provocative idea that EC can actively impact metabolism at the tissue level by bringing to light the dynamic contribution of EC metabolic activity to whole-body energy homeostasis during obesity. Although currently available tools do not provide the resolution required for a quantitative in vivo assessment of glucose consumption specifically by microvascular EC, our data strongly suggest that increasing EC metabolic activity leads to increased glucose uptake from the circulation and thus positively influences systemic glucose usage and tolerance. In addition, several pieces of available knowledge provide support for our hypothesis. First, EC are uniquely positioned at the interface between the bloodstream and the tissue parenchymal cells, which provides them with preferential access to circulating metabolic substrates. Second, glucose uptake by these cells is mediated via glucose transporter 1 (GLUT1), an insulin-independent transporter that is widely recognized to regulate basal glucose disposal. Interestingly, a direct connection between endothelial GLUT1 level and whole-organ glucose metabolism under physiological conditions was reported in a mouse model of EC-*Hif1a* depletion (*Huang et al., 2012*). Third, EC are among the most abundant cell types in the human body, accounting for 2–7% of the total cell number (*Bianconi et al., 2013*; *Sender et al., 2016*), with the majority of these EC residing within capillaries. Therefore, it is plausible that the

combined increase in EC metabolic activity with the expansion of capillary EC number can impact the overall systemic glucose homeostasis. The exact contribution of EC metabolism to whole-body substrate utilization, however, merits further investigation, as EC-FoxO1 depletion reprogrammed both metabolic activity and angiogenic responses of EC.

Beyond the remarkable effect that we observed on whole-body glucose metabolism, our findings support the notion that the substantial increase in vascular growth resulting from EC-FoxO1 depletion also impacted lipid handling under HF feeding conditions. In fact, HF-fed EC-FoxO1 KD mice displayed lower adiposity and serum levels of triglycerides and glycerol and less lipid accumulation in the liver. Although this could constitute a direct consequence of vascular growth, as fatty acid oxidation is used to support de novo nucleotide synthesis in EC (*Schoors et al., 2015*), it is also likely to involve secondary compensatory effects triggered by sustained imbalances in global glucose homeostasis that result from increased EC metabolic activity.

A number of previous reports have shown that lower expression of EC-*Foxo1* leads to disrupted vascular remodeling (*Furuyama et al., 2004*; *Sengupta et al., 2012*; *Dharaneeswaran et al., 2014*), which seemingly contradicts the phenotype we observed. A significant part of this discrepancy may arise from differences in experimental design and approach. Those studies used Cre-deleter models (Tie2-Cre and Cdh5-Cre) that broadly affect all vascular beds (including lymphatic endothelium) beginning at an early embryonic stage and also exhibit substantial Cre recombinase activity within hematopoietic lineages (*Chen et al., 2009*; *Tang et al., 2010*). In contrast, we employed a model of inducible EC-*Foxo1* depletion in adult mice, in which Cre activity was restricted to mature microvascular EC. Consistent with our findings, the induced depletion of EC-*Foxo1* in newborn mice resulted in increased EC growth and vessel enlargement within mouse retina (*Wilhelm et al., 2016*), although to a greater extent than the phenotype observed in our study. Interestingly, microvascular EC possess distinct tissue-specific molecular signatures (*Nolan et al., 2013*). In addition, it has been shown that EC-depletion of FoxO transcription factors can result in unique biological consequences in different tissue contexts (*Paik et al., 2007*). Thus, it is likely that both the developmental stage and the tissue microenvironment contribute substantially to phenotype observed with EC-*Foxo1* depletion. These study-specific features emphasize that the methodological details need to be carefully considered in the interpretation of data from EC-*Foxo1* depletion and highlight that the impact of EC-depletion cannot necessarily be extrapolated to different tissue contexts.

Altogether, our study reveals that EC-FoxO1 depletion evokes increased glycolytic capacity of endothelial cells and enables microvascular expansion in conditions where an angiogenic stimulus is present, as exemplified by the capillary expansion seen in white adipose depots in HF-fed mice. The repercussions of these combined influences include profound improvements in white adipose tissue capacity to cope with the stimulus of sustained nutrient excess and systemic enhancement of glucose clearance. In conclusion, these effects clearly define FoxO1 as the major regulator of the EC response to HF diet through the repression of beneficial metabolic and angiogenic adaptations in response to the stimulus of nutrient excess. Finally, this study brings to light an unappreciated role of EC as a distinctive metabolic entity rather than a simple exchange interface and highlights the modulation of endothelial metabolic and angiogenic activity as a potential target in the treatment of obesity-related disturbances.

# Materials and methods

**Key resources table**

| Reagent type (species) or resource | Designation | Source or reference | Identifiers | Additional information |
|---|---|---|---|---|
| Strain, strain background (M. musculus, FVB/n) | Foxo1[f/f] (FVB-Foxo1[tm1Rdp]) | From the laboratory of Dr Tara L. Haas | | Generated by crossing Foxo1,3,4f/f mice (FVB background; from the laboratory of Dr. Ronald A. DePinho) with FVB/n mice. |

*Continued on next page*

*Continued*

| Reagent type (species) or resource | Designation | Source or reference | Identifiers | Additional information |
|---|---|---|---|---|
| Strain, strain background (M. musculus, FVB/n) | Foxo1,3$^{f/f}$ (FVB-Foxo1$^{tm1Rdp}$; Foxo3$^{tm1Rdp}$) | From the laboratory of Dr Tara L. Haas | | Generated by crossing Foxo1,3,4f/f mice (FVB background; from the laboratory of Dr. Ronald A. DePinho) with FVB/n mice. |
| Strain, strain background (M. musculus, (C57BL/6 x CBA)F2) | Pdgfb-iCre [B6-Tg(Pdgfb-icre /ERT2)1Frut] | From the laboratory of Dr. Marcus Fruttiger | | The founder mouse was a kind gift from Dr. Marcus Fruttiger.. |
| Strain, strain background (M. musculus, FVB.B6) | Foxo1$^{iEC-D}$ [FVB.B6-Tg (Pdgfb-icre/ERT2) 1Frut; Foxo1$^{tm1Rdp}$] | This paper | | Mice were generated by cross-breeding Pdgfb-iCre with Foxo1$^{f/f}$ mice, followed by backcrossing offspring with Foxo1$^{f/f}$ mice for 3 + generations |
| Strain, strain background (M. musculus, FVB.B6) | Foxo1,3$^{iEC-D}$ [FVB.B6-Tg (Pdgfb-icre/ERT2) 1Frut; Foxo1$^{tm1Rdp}$; Foxo3$^{tm1Rdp}$] | This paper | | Mice were generated by cross-breeding Pdgfb-iCre with Foxo1,3$^{f/f}$ mice, followed by backcrossing offspring with Foxo1,3$^{f/f}$ mice for 3 + generations |
| Cell line (primary mouse adipose derived endothelial cells, male) | Endothelial cell fraction from adipose tissue | This paper | | Isolated freshly for each experiment |
| Cell line (primary mouse skeletal muscle endothelial cells, male) | Cultured microvascular EC, SMEC | This paper | | |
| Antibody | Biotin Rat Anti-Mouse CD31 | BD Pharmingen | 553371 | Cell purification |
| Antibody | MitoProfile Total OXPHOS Rodent WB Antibody Cocktail | Abcam | ab110413 | WB (1:500) |
| Antibody | Mouse Anti-β-actin | Santa Cruz Biotechnology | sc-47778 | WB (1:5000) |
| Antibody | Peroxidase AffiniPure Goat Anti-Mouse | Jackson ImmunoResearch | 115-035-003 | WB (1:10,000) |
| Antibody | Peroxidase AffiniPure Goat Anti-Rabbit | Jackson ImmunoResearch | 111-035-003 | WB (1:10,000) |
| Antibody | Purified Rat Anti-Mouse CD16/CD32 (Mouse BD Fc Block) | BD Pharmingen | 553141 | Cell purification |
| Antibody | Purified Rat Anti-Mouse CD144 | BD Pharmingen | 555289 | Cell purification |
| Antibody | Rabbit Anti-α/β-tubulin | Cell Signaling | 2148 | WB (1:1000) |

*Continued on next page*

*Continued*

| Reagent type (species) or resource | Designation | Source or reference | Identifiers | Additional information |
|---|---|---|---|---|
| Antibody | Rabbit Anti-Akt | Cell Signaling | 9272 | WB (1:1000) |
| Antibody | Rabbit Anti-HSL | Cell Signaling | 4107 | WB (1:1000) |
| Antibody | Rabbit Anti-Hexokinase | Cell Signaling | 2867T | WB (1:1000) |
| Antibody | Rabbit Anti-PFKFB3 | Cell Signaling | 13123S | WB (1:1000) |
| Antibody | Rabbit Anti-phospho-Akt (Ser473) | Cell Signaling | 4058 | WB (1:1000) |
| Antibody | Rabbit Anti- phospho-HSL (Ser563) | Cell Signaling | 4139 | WB (1:1000) |
| Sequence-based reagent (oligonucleotide) | oFK1ckA: GCT TAG AGC AGA GAT GTT CTC ACA TT | ThermoFisher Scientific | NA | |
| Sequence-based reagent (oligonucleotide) | oFK1ckB: CCA GAG TCT TTG TAT CAG GA AAT AA | ThermoFisher Scientific | NA | |
| Sequence-based reagent (oligonucleotide) | oFK1ckC: CAA GTC CAT TAA TTC AGC ACA TTG A | ThermoFisher Scientific | NA | |
| Sequence-based reagent (oligonucleotide) | oFK2ckA: ATT CCT TTG GAA ATC AAC AAA ACT | ThermoFisher Scientific | NA | |
| Sequence-based reagent (oligonucleotide) | oFK2ckB: TGC TTT GAT ACT ATT CCA CAA ACCC | ThermoFisher Scientific | NA | |
| Sequence-based reagent (oligonucleotide) | oFK1ckC: AGA TTT ATG TTC CCA CTT GCT TCCT | ThermoFisher Scientific | NA | |
| Peptide, recombinant protein | Humalog Insulin | Lilly | NA | |
| Commercial assay or kit | PureLink Genomic DNA Mini Kit | ThermoFisher Scientific | K182001 | |
| Commercial assay or kit | EnzyFluo™L-lactate Assay Kit | BioAssay Systems | EFLLC-100 | |
| Commercial assay or kit | Glycerol Assay Kit | Sigma-Aldrich | MAK117 | |
| Commercial assay or kit | Glucose (HK) Assay | Sigma-Aldrich | GAHK20 | |
| Commercial assay or kit | Lactate-Glo Assay | Promega | J5021 | |
| Commercial assay or kit | RNeasy Mini Kit | Qiagen | 74106 | |
| Commercial assay or kit | Triglyceride Colorimetric Assay kit | Cayman Chemical Company | 10010303 | |

*Continued*

| Reagent type (species) or resource | Designation | Source or reference | Identifiers | Additional information |
|---|---|---|---|---|
| Chemical compound, drug | AS1842856 FoxO1 inhibitor | EMD Millipore | 344355 | |
| Chemical compound, drug | Isoproterenol | Tocris | 1747 | |
| Chemical compound, drug | Tamoxifen | Sigma | T5648 | |
| Software, algorithm | Image J Analysis Software | National Institutes of Health | https://imagej.nih.gov/ij/download.html | |
| Software, algorithm | GraphPad Prism Version 6.07 | GraphPad Software Inc. | https://www.graphpad.com/scientific-software/prism/ | |
| Other | 11 kcal% fat w/ sucrose Surwit Diet | Research Diets | D12329 | |
| Other | 58 kcal% fat w/sucrose Surwit Diet | Research Diets | D12331 | |
| Other | BODIPY 493/503 | ThermoFisher Scientific | D3922 | |
| Other | Dynabeads | ThermoFisher Scientific | 14311D | |
| Other | Fast TaqMan Master Mix | ThermoFisher Scientific | 4444963 | |
| Other | M-MLV reverse transcriptase | New England Biolabs | M0253 | |
| Other | Streptavidin Particles Plus - DM | BD IMag | 557812 | |
| Other | Rhodamine labeled *Griffonia* (*Bandeiraea*) *Simplicifolia* lectin | VectorLabs | RL1102 | |
| Other | QIAzol Lysis Reagent | Qiagen | 79306 | |
| Other | SuperSignal West Pico | ThermoFisher Scientific | 34080 | |
| Other | Type I collagenase | ThermoFisher Scientific | 17100–017 | |
| Other | Type II collagenase | ThermoFisher Scientific | 17101–005 | |

## Mice

*Foxo1*$^{f/f}$ mice and *Foxo1,3*$^{f/f}$ mice were derived from outbreeding of *Foxo1,3,4*$^{f/f}$ mice (*Paik et al., 2007*) with wild-type FVB/n mice, and genotyped to ensure homozygosity of the floxed allele(s). To permit the inducible endothelial-specific manipulation of *Foxo1* and *Foxo3a*, these mice were bred with *Pdgfb*-iCreERT2 mice (C57Bl/6 background) (*Claxton et al., 2008*). Offspring were back-crossed with *Foxo1*$^{f/f}$ or *Foxo1,3*$^{f/f}$ founders for a minimum of 3 generations prior to experimentation to establish genotypes Cre⁻;*Foxo1*$^{f/f}$ and Cre⁺;*Foxo1*$^{f/f}$ or Cre⁻;*Foxo1,3*$^{f/f}$ and Cre⁺;*Foxo1,3*$^{f/f}$, respectively.

## Animal studies

We performed five separate animal studies using only male mice. To generate mice with postnatal endothelial cell-specific deletion of FoxO1 (EC-FoxO1 KD), or double deletion of *Foxo1* and *Foxo3*

(EC-FoxO1,3 KD), Cre-mediated recombination was induced in 4–6 weeks old Cre$^+$;$Foxo1^{f/f}$; mice and Cre$^+$;$Foxo1,3^{f/f}$; mice by five consecutive *i.p.* injections of 200 µL tamoxifen (15 mg/mL in corn oil). Recombination of *Foxo1* (and *Foxo3*) alleles was confirmed via PCR analysis of genomic DNA (*Paik et al., 2007*). In all experiments, EC-FoxO1 KD and EC-FoxO1,3 KD mice were compared with age-matched tamoxifen-injected Cre$^-$;$Foxo1^{f/f}$ or Cre$^-$;$Foxo1,3^{f/f}$ littermates (referred to as Control). No statistical method was used to predetermine the sample size. At 6–8 weeks of age, male mice within each litter were assigned randomly, according to their genotypes, to either normal chow (NC, 11% kcal from fat) or high-fat (HF, 58% kcal from fat, Surwit Diet) groups. Water and diet were provided *ad libitum*. Body weights and food intake of mice included in studies 1 and 2 were recorded weekly. Specific animal groups and the tests conducted were as followed: *Group 1:* Control and EC-FoxO1 KD mice received 16 week NC or HF diets (n = 6–7/group) and underwent body composition imaging at week 13, GTT and ITT (weeks 14, 15 respectively) and tissue collection at week 16 for whole tissue analyses (histology and RNA). Based on the apparent lack of influence of *Foxo1* deletion under NC diet in this group (see Results), subsequent testing focused on comparing the genotype differences observed in the HF condition: *Group 2:* Control and EC-FoxO1 KD mice (n = 7 and 8, respectively) underwent 14 week HF diet, metabolic testing (CLAMS) at week 13; ITT and tissue collection at week 14 for mitochondrial respiration, electron microscopy and serum analyses of TGs and glycerol. *Group 3 and 4*: Control and EC-FoxO1 KD mice (n = 5/group) and *Group 5:* Control and EC-FoxO1,3 KD (n = 6/group) underwent HF diet for 7, 5 and 14 weeks, respectively. Groups 3 and 5 were used to analyze 4 hr fasting glucose, and groups 3, 4 and 5 were used to isolate endothelial cell fractions from adipose depots for RNA and for glucose uptake, glucose consumption and lactate release assays. Animal studies were conducted in accordance with the American Physiological Society's guiding principles in the Care and Use of Animals, following protocols approved by the York University Committee on Animal Care.

## Body composition analysis

Body composition was examined by micro-computed tomography (Skyscan 1278; Bruker) using the step and shoot function, averaging four images/frame with a rotation step of 0.75 degrees. Mice were scanned at a voltage of 50kV, a current of 200 A with a 0.5 aluminum filter while anesthetized with isoflurane. Images were reconstructed using NRecon (local) and the entire trunk area fat mass was analyzed using CTAn.

## Whole-body metabolism analysis

Mice were monitored individually for oxygen consumption, carbon dioxide production, respiratory exchange ratio and locomotor activity using Indirect Calorimetry with the Columbus Comprehensive Lab Animal Monitoring System (CLAMS, Columbus Instruments, USA). Mice acclimated to CLAMS cages for 24 hr then data were recorded every 5 min for a 48 hr period. Mice had *ad libitum* access to food and water. $VO_2$ and $CO_2$ were normalized to body weight. Respiratory exchange ratio (RER) was calculated as the volume of $CO_2$ relative to the volume of oxygen ($VCO_2/VO_2$).

## Intra-peritoneal insulin and glucose tolerance tests (ITT and GTT)

For ITT, Control and EC-FoxO1 KD mice were fasted for 4 hr and received an *i.p.* injection of insulin (0.75 U/kg BW). GTTs were conducted in overnight-fasted control and EC-FoxO1 KD mice using an *i.p.* injection of glucose (1.75 g/kg BW). Blood glucose was measured by glucometer (Freestyle Lite, Abbott Diabetes Care, ON, Canada) at post-injection time-points 0, 20, 40 and 60 min (ITT) or 0, 30, 60, 90 and 120 min (GTT).

## In vivo insulin stimulation

Insulin stimulation of skeletal muscle was conducted as described previously (*Nwadozi et al., 2016*) and the phosphorylation state of Akt was assessed by Western blotting.

## Ex vivo adipose explant incubation

Epididymal fat pads were cut into ~ 80 mg pieces and pre-incubated with low glucose DMEM containing 1% fatty acid-free BSA for 30 min (37°C) before 30 min incubation with insulin (25 mU/mL) or

isoproterenol (10 μmol/L). Tissue explants were snap frozen in liquid nitrogen and the phosphorylation states of Akt or HSL were assessed by Western blotting.

## Mitochondrial respiration

Respirometry studies in freshly extracted epididymal white adipose tissue (eWAT) were performed using high-resolution respirometry (Oroboros Oxygraph-2k, Oroboros Instruments, Crop, Innsbruck, Austria). EWAT fat pads were prepared as done previously (*MacPherson et al., 2016*), minced in MIR05 buffer, weighed and immediately placed in separate Oroboros Oxygraph-2k in respiration media (MIR05) containing 20 mmol/L creatine (*Saks et al., 1995*). State three respiration was stimulated with the addition of ADP [5 mmol/L] in the presence of pyruvate [5 mmol/L] and malate [4 mmol/L] followed by glutamate [10 mmol/L] and succinate [20 mmol/L]. Respiration experiments were completed at 37°C before the oxygraph chamber reached 150 mmol/L [$O_2$]. Mitochondrial membrane integrity was tested with the addition of 10 mmol/L cytochrome c oxidase at the end of each protocol.

## Western Blot

Total protein extraction from isolated cells or tissues was performed as previously described (*Milkiewicz et al., 2011*). Primary antibodies were as follows: FoxO1, Ser473-pAkt, Akt, Ser563-pHSL, HSL, HK2, PFKFB3, α/β-tubulin, β-actin and Mitoprofile Total OXPHOS Cocktail. Secondary antibodies were goat anti-rabbit or anti-mouse IgG-horseradish peroxidase. Membranes were developed using enhanced chemiluminescence and densitometry analysis was performed with ImageJ Analysis Software (NIH).

## Imaging of adipose tissue

For microvascular quantification, pieces of eWAT were fixed in 4% formaldehyde and stained with BODIPY 493/503 (0.25 μg/mL) and rhodamine-*Griffonia Simplicifolia* lectin (1:100) to visualize adipocytes and microvessels, respectively. Images were taken with a Zeiss LSM700 confocal microscope (10x or 20x objectives) using identical gain settings for all samples. Microvascular content and branchpoint numbers were quantified from 3 to 4 10x fields of view per animal, and vessel diameters from 20x images, using Image J Analysis Software (NIH). For morphometric analysis of adipocytes, eWAT was fixed in 4% formaldehyde. Paraffin embedding, sectioning and hematoxylin and eosin staining were carried out by The Centre for Phenogenomics (Toronto, Canada). Adipocyte area was analyzed in three randomly selected fields of view (4x objective) using ImageJ Analysis Software (NIH). For quantification of capillaries, de-paraffinized sections were stained with fluorescein isothiocyanate-conjugated *Griffonia Simplicifolia* Lectin-1 (1:100) and Rhodamine Wheat Germ Agglutinin (1:200). Images were acquired using a 10X objective on a Zeiss inverted microscope equipped with a digital cooled CCD camera, capturing 3–4 independent fields of view per mouse. Image J software was used to quantify the numbers of capillaries and adipocytes in corresponding *Griffonia* and Wheat Germ Agluttinin-stained images, respectively.

## Muscle histology and Electron Microscopy (EM)

Cross-sections of EDL were stained with *Griffonia simplificolia*-FITC for assessment of capillary to fiber ratio (*Nwadozi et al., 2016*). Pieces of EDL muscles from HF-fed Control and EC-FoxO1 KD mice (n = 4/group) were fixed in 2% glutaraldehyde and 4% paraformaldehyde in 0.1M phosphate buffer (pH = 7.4) and sent for EM processing at the Hospital for Sick Children (Toronto, Canada). Cross-sectional images of capillaries (identified by lack of smooth muscle cell coverage) were captured by a blinded EM technician. Luminal diameters and endothelial cross-sectional areas of all detected capillaries within each sample were assessed using Image J software.

## Measurement of lipids and lactate

Serum triglycerides and glycerol levels were measured using commercial kits. Triglycerides were also measured in homogenates of liver and gastrocnemius muscle. Lactate levels were assessed in serum from HF-fed Control and EC-FoxO1 KD mice and in phenol-free cell culture medium of adipose-derived endothelial cells and SMEC also using a commercial kit.

## Endothelial cell isolation and culture

SMEC were isolated from collagenase digested skeletal muscle using rat anti-mouse VE-cadherin antibody-coated Dynabeads and biotinylated rat anti-mouse CD31 antibody-coated streptavidin-coupled beads, cultivated as described previously (*Roudier et al., 2013*), and used in experiments between passages 4 and 7. For glucose and FoxO1 inhibitor treatments, cells were plated and after overnight attachment, culture medium was replaced by low (5 mmol/L) or high-glucose (25 mmol/L) DMEM 10% FBS and incubated for 48 hr and 1 μmol/L AS1842856 was added to the medium 18 hr prior to testing. To examine the influence of FoxO1 inhibition on protein levels of glycolytic enzymes, SMEC were plated sparsely. After overnight attachment, culture medium (high-glucose DMEM with 10% FBS) was replaced by high-glucose DMEM with 0.1% FBS and the cells were stimulated with 1 μmol/L AS1842856 for 24 hr before lysis in RIPA buffer for protein extraction. For the isolation of adipose-derived endothelial cells, white adipose depots were pooled and digested with 0.5% Type I collagenase for 20 min at (37°C) with shaking. Centrifugation (300x*g* for 5 min) was used to separate adipocytes from the stromal vascular fraction (SVF). The re-suspended SVF was passed through a 100 μm-cell strainer, then pre-incubated with rat anti-mouse CD16/CD32 coupled to Dynabeads to deplete immune cells. Endothelial cells were selected using rat anti-mouse VE-cadherin antibody-coated Dynabeads and biotinylated rat anti-mouse CD31 antibody-coated streptavidin-coupled beads. Cells were plated on gelatin-coated plates and maintained in high-glucose DMEM (10% FBS) until utilization the following day.

## Metabolic assays

Glucose uptake was assessed in freshly adipose-derived endothelial cells and in SMEC using Glucose Uptake-Glo Assay following manufacturer's instructions. To examine the effects of *Foxo1* depletion on the glycolytic capacity of endothelial cells, freshly isolated adipose-derived endothelial cells were incubated with high-glucose DMEM plus 10% dialyzed FBS for 32 hr. The effects of FoxO1 inhibition were examined in SMEC pre-treated with 1 μmol/L AS1842856 for 24 hr in high-glucose DMEM plus 0.1% FBS following incubation with high-glucose DMEM plus 10% dialyzed FBS for 32 hr. Glucose consumption and lactate production were determined in cell culture medium at different time points (0, 8 and 24 hr) using commercial kits.

## Gene expression analysis

Total RNA was isolated from liver, skeletal muscle (plantaris), adipose tissue (BAT, eWAT and subcutaneous), adipose-derived endothelial cell and CD16/CD32 fractions and SMEC using RNeasy Mini Kit (Qiagen Inc.), reverse-transcribed and analyzed by real-time PCR on the Rotor-Gene Q platform (Qiagen Inc.) using Fast TaqMan Master Mix and TaqMan primer sets (listed in *Supplementary file 2*). Each target gene was calculated relative to *Hprt1* or *Actb* levels and expressed as $2^{-\Delta Ct}$.

## Data reporting and statistical analyses

All data reported are for independent biological replicates; each mouse being considered as one biological replicate. Averaged values were used when technical replicates (analysis of the same sample in duplicates) were performed. For in vitro assays, experiments were repeated at least three times and a sample size of $\geq 3$ biological replicates was used. Samples or data points were excluded only in the case of a technical equipment or human error that caused a sample to be poorly controlled. Statistical analyses were performed using Prism 5 (GraphPad Software Inc.). Significance was established at $p < 0.05$, by unpaired Student's *t*-test or 2-way ANOVA with Bonferroni post hoc tests, as appropriate. Data are shown as means $\pm$ SEM and *P* values are indicated in each Figure as *$p < 0.05$, **$p < 0.01$, ***$p < 0.001$.

## Acknowledgments

We thank T Gustafsson (Karolinska Institutet) for critical reading of the manuscript and valuable discussions. This work was funded by Canadian Institutes of Health Research Grant MOP-130491 (to TLH and ER) and Natural Science and Engineering Research Council of Canada (Discovery Grant #436138–2013 to CGRP). The funders had no role in study design, data collection and interpretation, or the decision to submit the work for publication.

## Additional information

### Funding

| Funder | Grant reference number | Author |
|---|---|---|
| Canadian Institutes of Health Research | MOP-130491 | Emilie Roudier<br>Tara L Haas |
| Natural Sciences and Engineering Research Council of Canada | Discovery Grant #436138-2013 | Christopher GR Perry |

The funders had no role in study design, data collection and interpretation, or the decision to submit the work for publication.

### Author contributions

Martina Rudnicki, Conceptualization, Data curation, Formal analysis, Validation, Investigation, Visualization, Methodology, Writing—original draft, Writing—review and editing; Ghoncheh Abdifarkosh, Emmanuel Nwadozi, Sofhia V Ramos, Data curation, Formal analysis, Investigation, Methodology, Writing—review and editing; Armin Makki, Diane M Sepa-Kishi, Data curation, Investigation, Methodology; Rolando B Ceddia, Resources, Investigation, Methodology, Writing—review and editing; Christopher GR Perry, Resources, Formal analysis, Funding acquisition, Investigation, Methodology, Writing—review and editing; Emilie Roudier, Conceptualization, Data curation, Formal analysis, Funding acquisition, Investigation, Methodology, Writing—review and editing; Tara L Haas, Conceptualization, Resources, Data curation, Formal analysis, Supervision, Funding acquisition, Validation, Investigation, Visualization, Writing—original draft, Project administration

### Author ORCIDs

Martina Rudnicki (iD) http://orcid.org/0000-0002-7863-5044
Emilie Roudier (iD) http://orcid.org/0000-0003-1014-6620
Tara L Haas (iD) http://orcid.org/0000-0001-8559-9574

### Ethics

Animal experimentation: All animal studies were conducted in strict accordance with the American Physiological Society's guiding principles in the Care and Use of Animals, following protocols approved by the York University Committee on Animal Care (reference number: 2017-19 R3).

### Decision letter and Author response

Decision letter https://doi.org/10.7554/eLife.39780.023
Author response https://doi.org/10.7554/eLife.39780.024

## Additional files

### Supplementary files

• Supplementary file 1. Supplementary Table 1. Tissue weights of Control and EC-FoxO1,3 KD mice after 14 weeks of HF diet
DOI: https://doi.org/10.7554/eLife.39780.019

• Supplementary file 2. TaqMan primer sets used in gene expression analysis
DOI: https://doi.org/10.7554/eLife.39780.020

• Transparent reporting form
DOI: https://doi.org/10.7554/eLife.39780.021

### Data availability

All data generated or analysed during this study are included in the manuscript. Individual replicates are plotted as dot plots, with the average and the appropriate error bars indicated for all numerical data.

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
