## [Decision Letter]

Thank you for submitting your article "Endothelial-specific *Foxo1* depletion prevents obesity-related disorders by increasing vascular metabolism and growth" for consideration by *eLife*. Your article has been reviewed by two peer reviewers, and the evaluation has been overseen by a Reviewing Editor and Mark McCarthy as the Senior Editor. The reviewers have opted to remain anonymous.

The reviewers have discussed the reviews with one another and the Reviewing Editor has drafted this decision to help you prepare a revised submission.

Summary:

This manuscript by Rudnicki and colleagues studies the impact of endothelial cell (EC)-specific depletion of FoxO1 on high-fat diet induced obesity. By utilizing a tamoxifen-inducible EC-specific *Foxo1* knock-out mouse model, the authors demonstrate that upon high-fat diet, EC *Foxo1* elimination results in greater vascular growth in visceral white adipose tissue and healthier adipose tissue expansion, resulting in improved glucose intolerance. The authors suggested that the mechanism for metabolic improvements is increased endothelial glycolytic capacity and enhanced endothelial proliferation. Overall, the present work provides novel insights into the role of endothelial FoxO1 on controlling vascular growth under diet induced obese conditions, particularly in adipose tissues.

Essential revisions:

A number of important issues would need to be clarified prior to additional review:

1) A major concern is that the authors have neglected or misinterpreted several important previous papers on FoxO1 depletion and endothelial cell function, including:

a) Furuyama T, et al., 2004, have shown that *Foxo1*-null mice are embryonic lethal at E11 due to impaired vasculogenesis;

b) The paper by Dharaneeswaran H, et al., 2014 demonstrated that EC-specific disruption of *Foxo1* reproduced the phenotypes of *Foxo1*-null mice;

c) Matsukawa M, et al. (Genes to Cells, 2009) also suggested that *Foxo1*-null endothelial cells showed an abnormal morphological response to VEGF-A and TGF-β, implicating an impairment in angiogenesis upon *Foxo1* depletion;

d) Even in the paper reported by Tanaka J, et al., 2009, which has been cited in the current manuscript, showed that EC-specific *Foxo1* knock-out mice displayed no gross or metabolic abnormalities;

e) The work done by Wilhelm K, et al., 2016, utilized the exact same mouse model as the authors have employed (*Pdgfb*-creERT2/*Foxo1* flox/flox) to demonstrate that EC loss of *Foxo1* caused hyperplastic vasculature but resulted in the inability of ECs to extend proper sprouts and uncoordinated vascular growth.

Considering many of these previously published articles are not supporting the findings in the present manuscript, the authors should carefully evaluate their own results and extensively discuss possible reasons for the inconsistency.

2) More functional evidence other than merely gene expression data is required to support the conclusion that EC depletion of *Foxo1* promotes endothelial glycolytic capacity, such as utilizing measurements of glycolytic rates on a Seahorse.

3) The authors state that EC-*Foxo1* depletion stimulates vascular growth during age-related adipose expansion (Figure 1) but no data are shown concerning the age-related adipose expansion (for example increased visceral adipose tissue mass normalized to body weight from week 0 to week 16 for each genetic background) and depletion of *foxo1* is shown only at mRNA level.

Similarly, all data are originated from direct comparison between two different genetic backgrounds (floxed mice are derived from outbreeding of *Foxo1*, 3, 4ff mice with wild-type FVB/n mice, described to be resistant to high fat diet-induced obesity, and the genetic background of *Pdgfb*-creERT2 mice is not indicated) under high fat diet. Comparisons between diets (normal chow and high fat) in the same genetic backgrounds (including body weights, fat depot and muscle weights as well as metabolic parameters) will be more relevant.

4) The lower adipocyte size described in Figure 4 may explain the greater microvascular content in Figure 2. The microvascular content must be expressed as capillary:adipocyte ratio as performed for muscle (Figure 3).

---

## [Author Response]

Essential revisions:A number of important issues would need to be clarified prior to additional review:1) A major concern is that the authors have neglected or misinterpreted several important previous papers on FoxO1 depletion and endothelial cell function, including:a) Furuyama T, et al., 2004, have shown that Foxo1-null mice are embryonic lethal at E11 due to impaired vasculogenesis;b) The paper by Dharaneeswaran H, et al., 2014 demonstrated that EC-specific disruption of Foxo1 reproduced the phenotypes of Foxo1-null mice;c) Matsukawa M, et al. (Genes to Cells, 2009) also suggested that Foxo1-null endothelial cells showed an abnormal morphological response to VEGF-A and TGF-β, implicating an impairment in angiogenesis upon Foxo1 depletion;d) Even in the paper reported by Tanaka J, et al., 2009, which has been cited in the current manuscript, showed that EC-specific Foxo1 knock-out mice displayed no gross or metabolic abnormalities;e) The work done by Wilhelm K, et al., 2016, utilized the exact same mouse model as the authors have employed (Pdgfb-creERT2/Foxo1 flox/flox) to demonstrate that EC loss of Foxo1 caused hyperplastic vasculature but resulted in the inability of ECs to extend proper sprouts and uncoordinated vascular growth.Considering many of these previously published articles are not supporting the findings in the present manuscript, the authors should carefully evaluate their own results and extensively discuss possible reasons for the inconsistency.

We respectfully disagree with the statement that we have misinterpreted several important previous papers on *Foxo1* depletion and endothelial cell function. We initially did not discuss findings from these papers for the sake of focus and brevity. However, we agree with the reviewers that the manuscript will benefit from added discussion of these points, which is included in the revised version of our manuscript. Importantly, we highlight a unifying emergent theme – that the influence of endothelial FoxO1 varies with context [developmental stage; tissue type; vessel type; presence of environmental stressors], and that these variations are unmasked through the use of a variety of experimental models and methodological approaches. We elaborate on this below.

Stage of development at the time of FoxO1 deletion in endothelial cells (EC) significantly influences outcomes:Whilst embryonic deletion of EC-*Foxo1* (or global FoxO1) leads to lethality with apparent disruption of vascular organization (not EC differentiation),^1-3^ post-natal conditional deletion of EC-FoxO1 is not lethal.^4^ These contrasting phenotypes (lethal vs. not lethal) observed with the genetic manipulation of FoxO1 in EC reflect the disparate incidence of angiogenesis in pre- vs. postnatal life and are also consistent with differences in the process of embryonic/developmental angiogenesis compared to angiogenesis within adult tissues. Throughout embryogenesis, the entire organism undergoes active expansion, and all vasculature (macro and micro) likewise is stimulated to expand, remodel and specialize according to location in the vascular tree. In turn, at the adult stage, angiogenesis occurs only when stimulated, and only within selective locations of the vasculature. Since endothelial cells (EC) are considered to be quiescent in the adult, angiogenic growth is orchestrated by a selected subset of mature EC, which do not exhibit the functional plasticity observed in embryonic EC. Moreover, recent data suggest that capillary formation and expansion in the adult occurs by clonal expansion of mature endothelial cells^5^ whereas random integration or cell mixing mediates developmental angiogenesis. All these differences ultimately impact how the vasculature expands at different stages. Importantly, the effects on cell morphology and sprouting and the poor responsiveness to VEGF and TGFβ in response to FoxO1 deletion reported by Furuyama et al.^1^ and Matsukawa et al.^6^ were observed with the use of embryonic stem cells, which exhibit some but not all characteristics of mature differentiated endothelial cells. These data were challenged by findings published by Paik et al.,^7^ who demonstrated that the deletion of FoxO transcription factors in EC of adult mice can either increase their proliferative capacity enhancing the response to VEGF or show no detectable phenotypic alteration. Thus, while findings reported with embryonic stem cells provide some clue of the effects of FoxO1 in EC, caution should be applied in extrapolating these results to the phenotype of mature EC. Furthermore, the differential processes involved in neovascularization within embryonic and adult vasculature offer a plausible rationale for the distinct phenotypes observed with FoxO1 deletion and limit the extent of direct comparison that can be made between the two types of experimental models.

The tissue microenvironment is crucial in establishing the consequence of EC-FoxO1 deletion: It has been reported that organ-specific differences remarkably affect phenotypic features of EC when depletion of EC-FoxO transcription factors is induced in adult mice,^7^ which is consistent with the current knowledge that microvascular EC show distinct molecular signatures within different tissues according to unique extravascular and intrinsic signals.^8^ Thus, the microenvironment exerts substantial control over angiogenesis,^9^ and tissues exhibit a range of angiogenic capacities at the adult stage. For example, in the absence of an added metabolic stimulus, skeletal muscle microvasculature in adult mice is very stable and angiogenic signals are minimal. Conversely, adipose tissue undergoes expansion through-out maturity as adipose depot size increases with age, and it produces a multitude of pro-angiogenic factors that help to coordinate capillary growth with tissue expansion. In this context and in agreement with the previous cited study,^7^ data from our current paper (and prior study,^10^ despite use of a different FoxO deletion strategy) provide strong evidence that these tissue environments affect the outcome of EC–*Foxo1* deletion. Indeed, while EC-*Foxo1* deletion exerted minimal phenotypic effects within skeletal muscle of normal chow-fed mice, increased vascular density was detectable in white adipose tissue of the same mice. In considering the study of Wilhelm et al.,^4^ it is apparent that we observed some similar and some disparate phenotypic features of EC with *Foxo1* depletion. Of note, although the authors had used a similar inducible deletion model, they induced the deletion of *Foxo1* in newborn mice and assessed the effects of endothelial-specific depletion of *Foxo1* specifically in retinal *developmental* angiogenesis. Consistent with their findings, we also observed more vessels and larger vessels in the adipose tissue vasculature when the deletion of *Foxo1* was induced in adult male mice. However, the more extensive disorganization of the vasculature observed by Wilhelm et al. but not in our study may be interpreted as the consequence of stronger stimulation of the EC by the potently pro-angiogenic environment within the maturing retina. Considering that those investigators closely tracked angiogenic processes within a narrow time window (i.e. 14-21 days) while we reported end-point vascular differences after 16 weeks of diet intervention, we cannot rule out the occurrence of alterations in endothelial cell sprouting/migration in our model. Our study was not designed to have the temporal resolution to assess those events, but it would be valuable to address this question in future investigations.

Differences between Cre deleter models used to elicit FoxO1 depletion: Importantly, the selection of Cre mouse model can lead to divergent phenotypes directly affecting the outcomes of EC-*Foxo1* depletion. Tie2-Cre and Cdh5-cre models used in Dharaneeswaran et al.^3^ and Tanaka et al.^11^ studies are widely employed to manipulate gene expression within the endothelium. However, both models show substantial Cre recombinase activity within hematopoietic lineages,^12-14^ which can confound the analysis and interpretation of the resultant phenotype. In addition, as both promoters are active in all endothelial cell populations, these constitutively active endothelial Cre transgenic models, affect all vascular beds, including lymphatic vasculature^15, 16^ beginning at an early embryonic stage being thus generally referred to as endothelial-specific knockout models. It is worth noting that in direct contradiction with previous studies,^1, 2^ Tanaka et al^11^ reported no lethality with embryonic EC-*Foxo1* deletion and as the reviewers also highlighted their EC-specific *Foxo1* knockout mice “were born at term in Mendelian ratios and showed no gross or metabolic abnormalities.” Strikingly, additional studies^2, 3^ that also used the Tie2-cre deleter to target Foxo1 reported embryonic lethality at approximately E11.0. The discrepancies between these studies and the Tanaka et al. study can probably be attributed to the variable cre activity associated with this Cre model. Indeed, Heffner et al. demonstrated that “embryos from the same litter display very different levels of Cre excision, ranging from endothelial-specific to nearly ubiquitous in non-target tissues.”^14^ In contrast with these previous studies, we did not use a constitutively active endothelial Cre transgenic model. Instead, the endothelial Cre/ERT2 model used in our study induces Cre recombinase activity at the adult stage (by nature of the timing of the tamoxifen-injection) under the control of the Platelet-derived growth factor B (*Pdgfb*) promoter.^17^*Pdgfb* expression is observed in capillary endothelial cells and in the endothelium of growing arteries during embryonic development and is restricted to capillaries into the adult stages.^18^ Therefore, the activation of *Pdgfb*-driven Cre recombinase activity at the adult stage favours the influence of genetic manipulation within capillaries not affecting all EC equally. As a consequence, the divergent pattern of Cre activity between our model and the Tie2 or Cadh5-deleter models^3, 11^ also could contribute to the differences in phenotype observed by us compared to those studies.

Overall, through these examples, we hope to have made a case that a) we do have a strong awareness of preceding studies and how their results relate to our own investigation and b) differences in the methodological approaches/experimental models need to be carefully considered in the interpretation of data from EC-*Foxo1* depletion when comparing the findings of these previously published articles and our data. Finally, we also would like to emphasize that ours is the first report documenting the effect of EC-*Foxo1* depletion in adult male mice, uniquely assessing angiogenesis of adipose tissue and skeletal muscle in the context of obesity. Overall, our study brings to light several novel concepts in vascular remodeling of these tissues in a pathophysiological context, which is relevant for understanding disease processes in the adult. While our answer to the reviewers has been extensive to justify our position, we have inserted a much briefer mention of these issues into the Discussion of the manuscript (seventh paragraph), as we share the reviewers’ point of view that it is beneficial to readers for us to explain differences in models.

2) More functional evidence other than merely gene expression data is required to support the conclusion that EC depletion of Foxo1 promotes endothelial glycolytic capacity, such as utilizing measurements of glycolytic rates on a Seahorse.

We have now added to the revised manuscript a new set of data to illustrate that EC-*Foxo1* depletion, as well as the inhibition of FoxO1 transcriptional activity, leads to increased glycolytic capacity of endothelial cells. For this experiment, we isolated microvascular endothelial cells from adipose tissue of adult male mice with EC-*Foxo1* depletion (EC-FoxO1 KD) fed a high-fat diet (HF) for 5 weeks. Due to the limited yield of cells obtained, we were not able to assess the glycolytic capacity of those cells measuring the extracellular acidification rate of cell medium by Seahorse analysis as suggested by the reviewers. As a surrogate approach to provide a functional read-out, we conducted matched measurements of glucose consumption and lactate production over time by measuring the lowering of glucose and the accumulation of lactate in the media over a 32 hour time window. In line with the data we provided before on fluorescent glucose uptake and lactate release, endothelial cells freshly isolated from adipose tissue of high fat-fed EC-FoxO1 KDmice displayed higher rates of glucose consumption than cells from floxed littermate controls. These findings were paralleled with increased lactate levels in the cell culture medium. These data argue in favor of increased glycolysis, the predominant bioenergetic pathway for ECs.^19^ To complement these observations, we repeated the measurement of lactate levels of microvascular endothelial cells from skeletal muscle cultivated in high-glucose conditions treated with FoxO1 inhibitor (AS1842856) with several protocol modifications. In the originally submitted manuscript, lactate levels were measured in cell medium after endothelial cells were incubated with FoxO1 inhibitor for only 18 hours in high-glucose medium containing 10% FBS. This time, we pre-incubated the cells for 24 hours with FoxO1 inhibitor in high-glucose medium with 0.1% FBS for 24 hours. After this 24h-pretreatment, culture medium was replaced by high-glucose DMEM plus 10% dialyzed FBS and cells were incubated in the presence and absence of FoxO1 inhibitor for 32 hours and cell media was collected in different time points (0, 8, 24 and 32 hours) for the assessment of glucose and lactate levels. Consistent with the findings in endothelial cells with EC-FoxO1 KD, we detected increased rates of glucose consumption and lactate productions when FoxO1 transcriptional activity was inhibited. Finally, we analyzed cultured microvascular EC treated with FoxO1 inhibitor for 24 hours for protein levels of hexokinase 2 and PFKFB3, and observed an increase in protein that matched with their elevated gene expression. We have added these new data to the revised manuscript (Figure 7D-E, Figure 8F-I, and Figure 8K-L).

3) The authors state that EC-Foxo1 depletion stimulates vascular growth during age-related adipose expansion (Figure 1) but no data are shown concerning the age-related adipose expansion (for example increased visceral adipose tissue mass normalized to body weight from week 0 to week 16 for each genetic background) and depletion of foxo1 is shown only at mRNA level.Similarly, all data are originated from direct comparison between two different genetic backgrounds (floxed mice are derived from outbreeding of Foxo1, 3, 4ff mice with wild-type FVB/n mice, described to be resistant to high fat diet-induced obesity, and the genetic background of Pdgfb-creERT2 mice is not indicated) under high fat diet. Comparisons between diets (normal chow and high fat) in the same genetic backgrounds (including body weights, fat depot and muscle weights as well as metabolic parameters) will be more relevant.

With respect to age-related adipose expansion: We thank for the reviewers for this observation. Because we did not conduct analyses of adipose depots at week 0 of the diet, we realize that we cannot directly comment on age-related adipose expansion. We have adjusted the wording in the Results and Discussion accordingly.

With respect to FoxO1 depletion: We have added Western blots of capillaries segments freshly isolated from skeletal muscle of EC-FoxO1 KD mice and Control littermates to provide further evidence of the efficacy of FoxO1 deletion at the protein level. Protein level decreased by ~ 70%, indicating that the decrease in FoxO1 mRNA is representative of a loss of FoxO1 protein (Figure 1D).

With respect to mouse background: We realize that our description of the mouse strains was not sufficiently detailed, leading to misinterpretation. *Foxo1^fl/fl^*mice are on an FVB background. The *Pdgfb*-iCreERT2 founder is on a C57BL/6 background. After cross-breeding these two mouse strains, subsequent generations were bred back to the *Foxo1^fl/fl^* strain to obtain homozygosity for the floxed alleles. Litters were back-crossed to the *Foxo1^fl/fl^* founder line for 3 generations prior to beginning experiments. It also is important to note that all experiments were conducted on littermates, sharing the identical genetic background. Littermate mice homozygous for the floxed *Foxo1* or *Foxo1,3* alleles but not expressing Cre recombinase were used as controls in our study. In our revised manuscript we have better described these details about the mice used in our study both in the Results and the Materials and methods section.

4) The lower adipocyte size described in Figure 4 may explain the greater microvascular content in Figure 2. The microvascular content must be expressed as capillary:adipocyte ratio as performed for muscle (Figure 3).

We understand the reviewers’ concern and agree that the adipocyte size could confound the assessment of microvascular density within adipose tissue. To address this question, we have quantified capillary number in paraffinized sections of eWAT from HF-fed Control and EC-FoxO1 KDmice (same experimental group used in the whole-mount staining analyses). The obtained data further confirm our previous findings of increased vascular density in adipose tissue of HF-fed EC-FoxO1 KD mice and clearly showed that the phenotype described in the originally submitted manuscript is not secondary to smaller adipocyte size. These data have been included in the revised manuscript (Figure 3F-G).

ReferenceList:

1) Furuyama T, Kitayama K, Shimoda Y, Ogawa M, Sone K, Yoshida-Araki K, Hisatsune H, Nishikawa S, Nakayama K, Nakayama K, Ikeda K, Motoyama N, Mori N. Abnormal angiogenesis in Foxo1 (Fkhr)-deficient mice. Journal of Biological Chemistry 2004 August 13;279(33):34741-9.

2) Sengupta A, Chakraborty S, Paik J, Yutzey KE, Evans-Anderson HJ. FoxO1 is required in endothelial but not myocardial cell lineages during cardiovascular development. Developmental Dynamics 2012 April;241(4):803-13.

3) Dharaneeswaran H, Abid MR, Yuan L, Dupuis D, Beeler D, Spokes KC, Janes L, Sciuto T, Kang PM, Jaminet SS, Dvorak A, Grant MA, Regan ER, Aird WC. FOXO1-mediated activation of Akt plays a critical role in vascular homeostasis. Circ Res 2014 July 7;115(2):238-51.

4) Wilhelm K, Happel K, Eelen G, Schoors S, Oellerich MF, Lim R, Zimmermann B, Aspalter IM, Franco CA, Boettger T, Braun T, Fruttiger M, Rajewsky K, Keller C, Bruning JC, Gerhardt H, Carmeliet P, Potente M. FOXO1 couples metabolic activity and growth state in the vascular endothelium. Nature 2016 January 14;529(7585):216-20.

5) Manavski Y, Lucas T, Glaser SF, Dorsheimer L, Gunther S, Braun T, Rieger MA, Zeiher AM, Boon RA, Dimmeler S. Clonal Expansion of Endothelial Cells Contributes to Ischemia-Induced Neovascularization. Circ Res 2018 March 2;122(5):670-7.

6) Matsukawa M, Sakamoto H, Kawasuji M, Furuyama T, Ogawa M. Different roles of Foxo1 and Foxo3 in the control of endothelial cell morphology. Genes Cells 2009 October;14(10):1167-81.

7) Paik JH, Kollipara R, Chu G, Ji H, Xiao Y, Ding Z, Miao L, Tothova Z, Horner JW, Carrasco DR, Jiang S, Gilliland DG, Chin L, Wong WH, Castrillon DH, DePinho RA. FoxOs are lineage-restricted redundant tumor suppressors and regulate endothelial cell homeostasis. Cell 2007 January 26;128(2):309-23.

8) Nolan DJ, Ginsberg M, Israely E, Palikuqi B, Poulos MG, James D, Ding BS, Schachterle W, Liu Y, Rosenwaks Z, Butler JM, Xiang J, Rafii A, Shido K, Rabbany SY, Elemento O, Rafii S. Molecular signatures of tissue-specific microvascular endothelial cell heterogeneity in organ maintenance and regeneration. Dev Cell 2013 July 29;26(2):204-19.

9) Augustin HG, Koh GY. Organotypic vasculature: From descriptive heterogeneity to functional pathophysiology. Science 2017 August 25;357(6353).

10) Nwadozi E, Roudier E, Rullman E, Tharmalingam S, Liu HY, Gustafsson T, Haas TL. Endothelial FoxO proteins impair insulin sensitivity and restrain muscle angiogenesis in response to a high-fat diet. FASEB Journal 2016 September;30(9):3039-52.

11) Tanaka J, Qiang L, Banks AS, Welch CL, Matsumoto M, Kitamura T, Ido-Kitamura Y, DePinho RA, Accili D. Foxo1 links hyperglycemia to LDL oxidation and endothelial nitric oxide synthase dysfunction in vascular endothelial cells. Diabetes 2009 October;58(10):2344-54.

12) Chen MJ, Yokomizo T, Zeigler BM, Dzierzak E, Speck NA. Runx1 is required for the endothelial to haematopoietic cell transition but not thereafter. Nature 2009 February 12;457(7231):887-91.

13) Tang Y, Harrington A, Yang X, Friesel RE, Liaw L. The contribution of the Tie2+ lineage to primitive and definitive hematopoietic cells. Genesis 2010 September;48(9):563-7.

14) Heffner CS, Herbert PC, Babiuk RP, Sharma Y, Rockwood SF, Donahue LR, Eppig JT, Murray SA. Supporting conditional mouse mutagenesis with a comprehensive cre characterization resource. Nat Commun 2012;3:1218.

15) Alva JA, Zovein AC, Monvoisin A, Murphy T, Salazar A, Harvey NL, Carmeliet P, Iruela-Arispe ML. VE-Cadherin-Cre-recombinase transgenic mouse: a tool for lineage analysis and gene deletion in endothelial cells. Dev Dyn 2006 March;235(3):759-67.

16) Srinivasan RS, Dillard ME, Lagutin OV, Lin FJ, Tsai S, Tsai MJ, Samokhvalov IM, Oliver G. Lineage tracing demonstrates the venous origin of the mammalian lymphatic vasculature. Genes Dev 2007 October 1;21(19):2422-32.

17) Claxton S, Kostourou V, Jadeja S, Chambon P, Hodivala-Dilke K, Fruttiger M. Efficient, inducible Cre-recombinase activation in vascular endothelium. Genesis 2008 February;46(2):74-80.

18) Hellstrom M, Kalen M, Lindahl P, Abramsson A, Betsholtz C. Role of PDGF-B and PDGFR-β in recruitment of vascular smooth muscle cells and pericytes during embryonic blood vessel formation in the mouse. Development 1999 June;126(14):3047-55.

19) De Bock K, Georgiadou M, Schoors S, Kuchnio A, Wong BW, Cantelmo AR, Quaegebeur A, Ghesquiere B, Cauwenberghs S, Eelen G, Phng LK, Betz I, Tembuyser B, Brepoels K, Welti J, Geudens I, Segura I, Cruys B, Bifari F, Decimo I, Blanco R, Wyns S, Vangindertael J, Rocha S, Collins RT, Munck S, Daelemans D, Imamura H, Devlieger R, Rider M, Van Veldhoven PP, Schuit F, Bartrons R, Hofkens J, Fraisl P, Telang S, Deberardinis RJ, Schoonjans L, Vinckier S, Chesney J, Gerhardt H, Dewerchin M, Carmeliet P. Role of PFKFB3-driven glycolysis in vessel sprouting. Cell 2013 August 1;154(3):651-63.